# Efficient sampling and noisy decisions

**Joseph A Heng[1], Michael Woodford[2], Rafael Polania[1]\***

[1]Department of Health Sciences and Technology, Federal Institute of Technology (ETH), Zurich, Switzerland; [2]Department of Economics, Columbia University, New York, United States

**Abstract** Human decisions are based on finite information, which makes them inherently imprecise. But what determines the degree of such imprecision? Here, we develop an efficient coding framework for higher-level cognitive processes in which information is represented by a finite number of discrete samples. We characterize the sampling process that maximizes perceptual accuracy or fitness under the often-adopted assumption that full adaptation to an environmental distribution is possible, and show how the optimal process differs when detailed information about the current contextual distribution is costly. We tested this theory on a numerosity discrimination task, and found that humans efficiently adapt to contextual distributions, but in the way predicted by the model in which people must economize on environmental information. Thus, understanding decision behavior requires that we account for biological restrictions on information coding, challenging the often-adopted assumption of precise prior knowledge in higher-level decision systems.

*'We rarely know the statistics of the messages completely, and our knowledge may change . . . what is redundant today was not necessarily redundant yesterday.' Barlow, 2001*.

## Introduction

It has been suggested that the rules guiding behavior are not arbitrary, but follow fundamental principles of acquiring information from environmental regularities in order to make the best decisions. Moreover, these principles should incorporate strategies of information coding in ways that minimize the costs of inaccurate decisions given biological constraints on information acquisition, an idea known as efficient coding (*Attneave, 1954*; *Barlow, 1961*; *Niven and Laughlin, 2008*; *Sharpee et al., 2014*). While early applications of efficient coding theory have primarily been to early stages of sensory processing (*Laughlin, 1981*; *Ganguli and Simoncelli, 2014*; *Wei and Stocker, 2015*), it is worth considering whether similar principles may also shape the structure of internal representations of higher-level concepts, such as the perceptions of value that underlie economic decision making (*Louie and Glimcher, 2012*; *Polanía et al., 2019*; *Rustichini et al., 2017*). In this work, we contribute to the efficient coding framework applied to cognition and behavior in several respects.

A first aspect concerns the range of possible internal representation schemes that should be considered feasible, which determines the way in which greater precision of discrimination in one part of the stimulus space requires less precision of discrimination elsewhere. Implementational architectures proposed in previous work assume a population coding scheme in which different neurons have distinct 'preferred' stimuli (*Ganguli and Simoncelli, 2014*; *Wei and Stocker, 2015*). While this is clearly relevant for some kinds of low-level sensory features such as orientation, it is not obvious that this kind of internal representation is used in representing higher-level concepts such as economic values. We instead develop an efficient coding theory for a case in which an extensive magnitude (something that can be described by a larger or smaller number) is represented by a set of

**\*For correspondence:**
rafael.polania@hest.ethz.ch

**Competing interests:** The authors declare that no competing interests exist.

processing units that 'vote' in favor of the magnitude being larger rather than small. The internal representation therefore necessarily consists of a finite collection of binary signals.

Our restriction to representations made up of binary signals is in conformity with the observation that neural systems at many levels appear to transmit information via discrete stochastic events (*Schreiber et al., 2002*; *Sharpee, 2017*). Moreover, cognitive models with this general structure have been argued to be relevant for higher-order decision problems such as value-based choice. For example, it has been suggested that the perceived values of choice options are constructed by acquiring samples of evidence from memory regarding the emotions evoked by the presented items (*Shadlen and Shohamy, 2016*). Related accounts suggest that when a choice must be made between alternative options, information is acquired via discrete samples of information that can be represented as binary responses (e.g., 'yes/no' responses to queries) (*Norman, 1968*; *Weber and Johnson, 2009*). The seminal *decision by sampling* (DbS) theory (*Stewart et al., 2006*) similarly posits an internal representation of magnitudes relevant to a decision problem by tallies of the outcomes of a set of binary comparisons between the current magnitude and alternative values sampled from memory. The architecture that we assume for imprecise internal representations has the general structure of proposals of these kinds; but we go beyond the above-mentioned investigations, in analyzing what an efficient coding scheme consistent with our general architecture would be like.

A second aspect concerns the objective for which the encoding system is assumed to be optimized. Information maximization theories (*Laughlin, 1981*; *Ganguli and Simoncelli, 2014*; *Wei and Stocker, 2015*) assume that the objective should be maximal mutual information between the true stimulus magnitude and the internal representation. While this may be a reasonable assumption in the case of early sensory processing, it is less obvious in the case of circuits involved more directly in decision making, and in the latter case an obvious alternative is to ask what kind of encoding scheme will best serve to allow accurate decisions to be made. In the theory that we develop here, our primary concern is with encoding schemes that maximize a subject's probability of giving a correct response to a binary decision. However, we compare the coding rule that would be optimal from this standpoint to one that would maximize mutual information, or to one that would maximize the expected value of the chosen item.

Third, we extend our theory of efficient coding to consider not merely the nature of an efficient coding system for a single environmental frequency distribution assumed to be permanently relevant — so that there has been ample time for the encoding rule to be optimally adapted to that distribution of stimulus magnitudes — but also an efficient approach to adjusting the encoding as the environmental frequency distribution changes. Prior discussions of efficient coding have often considered the optimal choice of an encoding rule for a single environmental frequency distribution that is assumed to represent a permanent feature of the natural environment (*Laughlin, 1981*; *Ganguli and Simoncelli, 2014*). Such an approach may make sense for a theory of neural coding in cortical regions involved in early-stage processing of sensory stimuli, but is less obviously appropriate for a theory of the processing of higher-level concepts such as economic value, where the idea that there is a single permanently relevant frequency distribution of magnitudes that may be encountered is doubtful.

A key goal of our work is to test the relevance of these different possible models of efficient coding in the case of numerosity discrimination. Judgments of the comparative numerosity of two visual displays provide a test case of particular interest given our objectives. On the one hand, a long literature has argued that imprecision in numerosity judgments has a similar structure to psychophysical phenomena in many low-level sensory domains (*Nieder and Dehaene, 2009*; *Nieder and Miller, 2003*). This makes it reasonable to ask whether efficient coding principles may also be relevant in this domain. At the same time, numerosity is plainly a more abstract feature of visual arrays than low-level properties such as local luminosity, contrast, or orientation, and therefore can be computed only at a later stage of processing. Moreover, processing of numerical magnitudes is a crucial element of many higher-level cognitive processes, such as economic decision making; and it is arguable that many rapid or intuitive judgments about numerical quantities, even when numbers are presented symbolically, are based on an 'approximate number system' of the same kind as is used in judgments of the numerosity of visual displays (*Piazza et al., 2007*; *Nieder and Dehaene, 2009*). It has further been argued that imprecision in the internal representation of numerical magnitudes may underly imprecision and biases in economic decisions (*Khaw et al., 2020*; *Woodford, 2020*).

It is well-known that the precision of discrimination between nearby numbers of items decreases in the case of larger numerosities, in approximately the way predicted by *Weber's Law*, and this is often argued to support a model of imprecise coding based on a logarithmic transformation of the true number (*Nieder and Dehaene, 2009*; *Nieder and Miller, 2003*). However, while the precision of internal representations of numerical magnitudes is arguably of great evolutionary relevance (*Butterworth et al., 2018*; *Nieder, 2020*), it is unclear why a specifically logarithmic transformation of number information should be of adaptive value, and also whether the same transformation is used independent of context (*Pardo-Vazquez et al., 2019*; *Brus et al., 2019*). Here, we report new experimental data on numerosity discrimination by human participants, where we find that our data are most consistent with an efficient coding theory for which the performance measure is the frequency of correct comparative judgments, and where people economize on the costs associated to learn about the statistics of the environment.

## Results

### A general efficient sampling framework

We consider a situation in which the objective magnitude of a stimulus with respect to some feature can be represented by a quantity $v$. When the stimulus is presented to an observer, it gives rise to an imprecise representation $r$ in the nervous system, on the basis of which the observer produces any required response. The internal representation $r$ can be stochastic, with given values being produced with conditional probabilities $p(r|v)$ that depend on the true magnitude. Here, we are more specifically concerned with discrimination experiments, in which two stimulus magnitudes $v_1$ and $v_2$ are presented, and the subject must choose which of the two is greater. We suppose that each magnitude $v_i$ has an internal representation $r_i$, drawn independently from a distribution $p(r_i|v_i)$ that depends only on the true magnitude of that individual stimulus. The observer's choice must be based on a comparison of $r_1$ with $r_2$.

One way in which the cognitive resources recruited to make accurate discriminations may be limited is in the variety of distinct internal representations that are possible. When the complexity of feasible internal representations is limited, there will necessarily be errors in the identification of the greater stimulus magnitude in some cases, even assuming an optimal decoding rule for choosing the larger stimulus on the basis of $r_1$ and $r_2$. One can then consider alternative encoding rules for mapping objective stimulus magnitudes to feasible internal representations. The answer to this efficient coding problem generally depends on the prior distribution $f(v)$ from which the different stimulus magnitudes $v_i$ are drawn. The resources required for more precise internal representations of individual stimuli may be economized with respect to either or both of two distinct cognitive costs. The first goal of this work is to distinguish between these two types of efficiency concerns.

One question that we can ask is wheter the observed behavioral responses are consistent with the hypothesis that the conditional probabilities $p(r|v)$ are well-adapted to the particular frequency distribution of stimuli used in the experiment, suggesting an efficient allocation of the limited encoding neural resources. The assumption of full adaptation is typically adopted in efficient coding formulations of early sensory systems (*Laughlin, 1981*; *Wei and Stocker, 2017*), and also more recently in applications of efficient coding theories in value-based decisions (*Louie and Glimcher, 2012*; *Polanía et al., 2019*; *Rustichini et al., 2017*).

There is also a second cost in which it may be important to economize on cognitive resources. An efficient coding scheme in the sense described above economizes on the resources used to represent each individual new stimulus that is encountered; however, the encoding and decoding rules are assumed to be precisely optimized for the specific distribution $f(v)$ of stimuli that characterizes the experimental situation. In practice, it will be necessary for a decision maker to learn about this distribution in order to encode and decode individual stimuli in an efficient way, on the basis of experience with a given context. In this case, the relevant design problem should not be conceived as choosing conditional probabilities $p(r|v)$ once and for all, with knowledge of the prior distribution $f(v)$ from which $v$ will be drawn. Instead, it should be to choose a rule that specifies how the probabilities $p(r|v)$ should adapt to the distribution of stimuli that have been encountered in a given context. It then becomes possible to consider how well a given learning rule economizes on the degree of information about the distribution of magnitudes associated with one's current context that is

required for a given level of average performance across contexts. This issue is important not only to reduce the cognitive resources required to implement the rule in a given context (by not having to store or access so detailed a description of the prior distribution), but in order to allow faster adaptation to a new context when the statistics of the environment can change unpredictably (*Młynarski and Hermundstad, 2019*).

## Coding architecture

We now make the contrast between these two types of efficiency more concrete by considering a specific architecture for internal representations of sensory magnitudes. We suppose that the representation $r_i$ of a given stimulus will consist of the output of a finite collection of $n$ processing units, each of which has only two possible output states ('high' or 'low' readings), as in the case of a simple perceptron. The probability that each of the units will be in one output state or the other can depend on the stimulus $v_i$ that is presented. We further restrict the complexity of feasible encoding rules by supposing that the probability of a given unit being in the 'high' state must be given by some function $\theta(v_i)$ that is the same for each of the individual units, rather than allowing the different units to coordinate in jointly representing the situation in some more complex way. We argue that the existence of multiple units operating in parallel effectively allows multiple repetitions of the same 'experiment', but does not increase the complexity of the kind of test that can be performed. Note that we do not assume any unavoidable degree of stochasticity in the functioning of the individual units; it turns out that in our theory, it will be efficient for the units to be stochastic, but we do not assume that precise, deterministic functioning would be infeasible. Our resource limits are instead on the number of available units, the degree of differentiation of their output states, and the degree to which it is possible to differentiate the roles of distinct units.

Given such a mechanism, the internal representation $r_i$ of the magnitude of an individual stimulus $v_i$ will be given by the collection of output states of the $n$ processing units. A specification of the function $\theta(v)$ then implies conditional probabilities for each of the $2^n$ possible representations. Given our assumption of a symmetrical and parallel process, the number $k_i$ of units in the 'high' state will be a sufficient statistic, containing all of the information about the true magnitude $v_i$ that can be extracted from the internal representation. An optimal decoding rule will therefore be a function only of $k_i$, and we can equivalently treat $k_i$ (an integer between 0 and $n$) as the internal representation of the quantity $v_i$. The conditional probabilities of different internal representations are then

$$p(k_i|v_i) = \binom{n}{k} \theta(v_i)^{k_i} (1 - \theta(v_i))^{n-k_i}. \tag{1}$$

The efficient coding problem for a given environment, specified by a particular prior distribution $f(v)$, will be to choose the encoding rule $\theta(v)$ so as to allow an overall distribution of responses across trials that will be as accurate as possible (according to criteria that we will elaborate further below). We can further suppose that each of the individual processing units is a threshold unit, that produces a 'high' reading if and only if the value $v_i - \eta_i$ exceeds some threshold $\tau$, where $\eta_i$ is a random term drawn independently on each trial from some distribution $f_\eta$ (*Figure 1*). The encoding function $\theta(v)$ can then be implemented by choosing an appropriate distribution $f_\eta$. This implementation requires that $\theta(v)$ be a non-decreasing function, as we shall assume.

## Limited cognitive resources

One measure of the cognitive resources required by such a system is the number $n$ of processing units that must produce an output each time an individual stimulus $v_i$ is evaluated. We can consider the optimal choice of $f_\eta$ in order to maximize, for instance, average accuracy of responses in a given environment $f(v)$, in the case of any bound $n$ on the number of units that can be used to represent each stimulus. But we can also consider the amount of information about the distribution $f(v)$ that must be used in order to decide how to encode a given stimulus $v_i$. If the system is to be able to adapt to changing environments, it must determine the value of $\theta$ (the probability of a 'high' reading) as a function of both the current $v_i$ and information about the distribution $f$, in a way that must now be understood to apply across different potential contexts. This raises the issue of how precisely the distribution $f$ associated with the current context is represented for purposes of such a

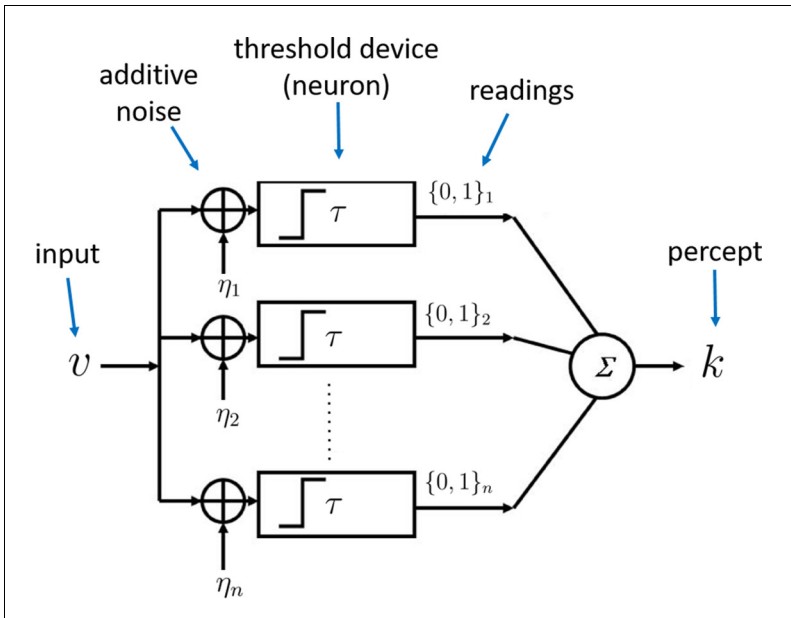

**Figure 1.** Architecture of the sampling mechanism. Each processing unit receives noisy versions of the input $v$, where the noisy signals are i.i.d. additive random signals independent of $v$. The output of the neuron for each sample is 'high' (one) reading if $v - \eta > \tau$ and zero otherwise. The noisy percept of the input is simply the sum of the outputs of each sample given by $k$.

calculation. A more precise representation of the prior (allowing greater sensitivity to fine differences in priors) will presumably entail a greater resource cost or very long adaptation periods.

We can quantify the precision with which the prior $f$ is represented by supposing that it is represented by a finite sample of $m$ independent draws $\tilde{v}_1, \ldots, \tilde{v}_m$ from the prior (or more precisely, from the set of previously experienced values, an empirical distribution that should after sufficient experience provide a good approximation to the true distribution). We further assume that an independent sample of $m$ previously experienced values is used by each of the processing units (*Figure 1*). Each of the $n$ individual processing units is then in the 'high' state with probability $\theta(v_i; \tilde{v}_1, \ldots, \tilde{v}_m)$. The complete internal representation of the stimulus $v_i$ is then the collection of $n$ independent realizations of this binary-valued random variable. We may suppose that the resource cost of an internal representation of this kind is an increasing function of both $n$ and $m$.

This allows us to consider an efficient coding meta-problem in which for any given values $(n, m)$ the function $\theta(v_i; \tilde{v}_1, \ldots, \tilde{v}_m)$ is chosen so as to maximize some measure of average perceptual accuracy, where the average is now taken not only over the entire distribution of possible $v_i$ occurring under a given prior $f(v)$, but over some range of different possible priors for which the adaptive coding scheme is to be optimized. We wish to consider how each of the two types of resource constraint (a finite bound on $n$ as opposed to a finite bound on $m$) affects the nature of the predicted imprecision in internal representations, under the assumption of a coding scheme that is efficient in this generalized sense, and then ask whether we can tell in practice how tight each of the resource constraints appears to be.

## Efficient sampling for a known prior distribution

We first consider efficient coding in the case that there is no relevant constraint on the size of $m$, while $n$ instead is bounded. In this case, we can assume that each time an individual stimulus $v_i$ must be encoded, a large enough sample of prior values is used to allow accurate recognition of the distribution $f(v)$, and the problem reduces to a choice of a function $\theta(v)$ that is optimal for each possible prior $f(v)$.

### Maximizing mutual information

The nature of the resource-constrained problem to be optimized depends on the performance measure that we use to determine the usefulness of a given encoding scheme. A common assumption in the literature on efficient coding has been that the encoding scheme maximizes the mutual information between the true stimulus magnitude and its internal representation (*Ganguli and Simoncelli, 2014*; *Polanía et al., 2019*; *Wei and Stocker, 2015*). We start by characterizing the optimal $\theta(v)$ for a given prior distribution $f(v)$, according to this criterion. It can be shown that for large $n$, the mutual information between $\theta$ and $k$ (hence the mutual information between $v$ and $k$) is maximized if the prior distribution $\hat{f}$ over $\theta$ is Jeffreys' prior (*Clarke and Barron, 1994*)

$$\hat{f}(\theta) = \frac{1}{\pi \sqrt{\theta(1-\theta)}}, \qquad (2)$$

also known as the arcsine distribution. Hence, the mapping $\theta(v)$ induces a prior distribution $\hat{f}$ over $\theta$

given by the arcsine distribution (*Figure 2a*, right panel). Based on this result, it can be shown that the optimal encoding rule $\theta(v)$ that guarantees maximization of mutual information between the random variable $v$ and the noisy encoded percept $k$ is given by (see Appendix 1)

$$\theta(v) = \left[\sin\left(\frac{\pi}{2}F(v)\right)\right]^2, \tag{3}$$

where $F(v)$ is the CDF of the prior distribution $f(v)$.

## Accuracy maximization for a known prior distribution

So far, we have derived the optimal encoding rule to maximize mutual information. However, one may ask what the implications are of such a theory for discrimination performance. This is important to investigate given that achieving channel capacity does not necessarily imply that the goals of the organism are also optimized (*Park and Pillow, 2017*). Independent of information maximization assumptions, here, we start from scratch and investigate what are the necessary conditions for minimizing discrimination errors given the resource-constrained problem considered here. We solve this problem for the case of two alternative forced choice tasks, where the average probability of error is given by (see Appendix 2)

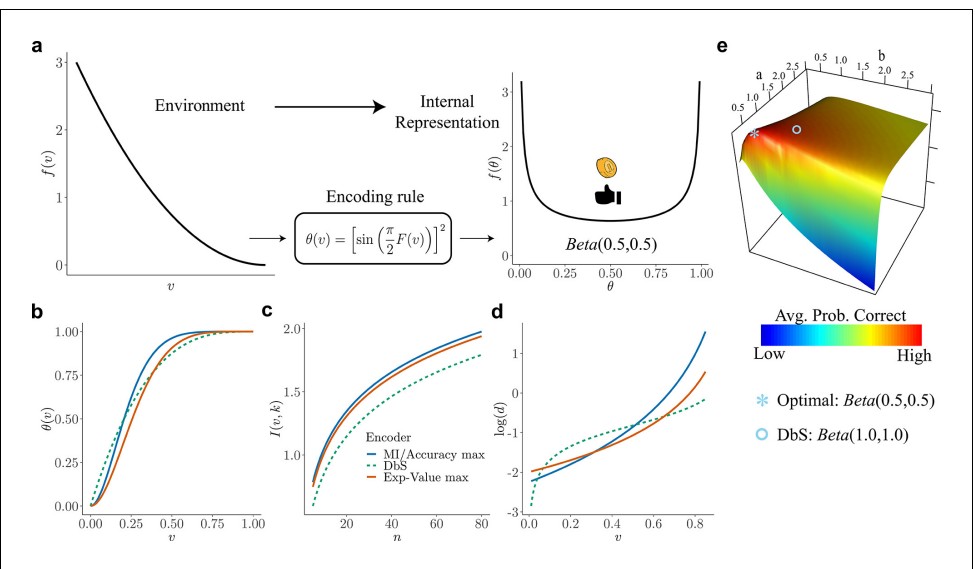

**Figure 2.** Overview of our theory and differences in encoding rules. (a) Schematic representation of our theory. Left: example prior distribution $f(v)$ of values $v$ encountered in the environment. Right: Prior distribution in the encoder space (*Equation 2*) due to optimal encoding (*Equation 3*). This optimal mapping determines the probability $\theta$ of generating a 'high' or 'low' reading. The ex-ante distribution over $\theta$ that guarantees maximization of mutual information is given by the arcsine distribution (*Equation 2*). (b) Encoding rules $\theta(v)$ for different decision strategies under binary sampling coding: accuracy maximization (blue), reward maximization (red), DbS (green dashed). (c) Mutual information $I(v,k)$ for the different encoding rules as a function of the number of samples $n$. As expected $I(v,k)$ increases with $n$, however the rule that results in the highest loss of information is DbS. (d) Discriminability thresholds $d$ (log-scaled for better visualization) for the different encoding rules as a function of the input values $v$ for the prior $f(v)$ given in panel a. (e) Graphical representation of the perceptual accuracy optimization landscape. We plot the average probability of correct responses for the large-$n$ limit using as benchmark a Beta distribution with parameters $a$ and $b$. The blue star shows the average error probability assuming that $f(\theta)$ is the arcsine distribution (*Equation 2*), which is the optimal solution when the prior distribution $f$ in known. The blue open circle shows the average error probability based on the encoding rule assumed in DbS, which is located near the optimal solution. Please note that when formally solving this optimization problem, we did not assume a priori that the solution is related to the beta distribution. We use the beta distribution in this figure just as a benchmark for visualization. Detailed comparison of performance for finite $n$ samples is presented in Appendix 7.

$$\mathrm{E}[\mathrm{error}] = \iint \mathrm{P}_{\mathrm{error}}[\theta(v_1), \theta(v_2)]\hat{f}(\theta_1)\hat{f}(\theta_2)\, d\theta_1 d\theta_2, \tag{4}$$

where $\mathrm{P}_{\mathrm{error}}[]$ represents the probability of erroneously choosing the alternative with the lowest value $v$ given a noisy percept $k$ (assuming that the goal of the organism in any given trial is to choose the alternative with the highest value). Here, we want to find the density function $\hat{f}(\theta)$ that guarantees the smallest average error (*Equation 4*). The solution to this problem is (Appendix 2)

$$\hat{f}(\theta) = \frac{1}{\pi\sqrt{\theta(1-\theta)}}, \tag{5}$$

which is exactly the same prior density function over $\theta$ that maximizes mutual information (*Equation 2*). Crucially, please note that we have obtained this expression based on minimizing the frequency of erroneous choices and not the maximization of mutual information as a goal in itself. This provides a further (and normative) justification for why maximizing mutual information under this coding scheme is beneficial when the goal of the agent is to minimize discrimination errors (i.e., maximize accuracy).

## Optimal noise for a known prior distribution

Based on the coding architecture presented in *Figure 1*, the optimal encoding function $\theta(v)$ can then be implemented by choice of an appropriate distribution $f_\eta$. It can be shown that discrimination performance can be optimized by finding the optimal noise distribution $f_\eta$ (Appendix 3) (*McDonnell et al., 2007*)

$$f_\eta(v) = \frac{\pi}{2}\sin[\pi(1 - F(\tau - v))]f(\tau - v). \tag{6}$$

Remarkably, this result is independent of the number of samples $n$ available to encode the input variable, and generalizes to any prior distribution $f$ (recall that $F$ is defined as its cumulative density function).

This result reveals three important aspects of neural function and decision behavior: First, it makes explicit why a system that evolved to code information using a coding scheme of the kind assumed in our framework must be necessarily noisy. That is, we do not attribute the randomness of peoples' responses to a particular set of stimuli or decision problem to unavoidable randomness of the hardware used to process the information. Instead, the relevant constraints are assumed to be the limited set of output states for each neuron, the limited number of neurons, and the requirement that the neurons operate in parallel (so that each one's output state must be statistically independent of the others, conditional on the input stimulus). Given these constraints, we show that it is efficient for the operation of the neurons to be random. Second, it shows how the nervous system may take advantage of these noisy properties by reshaping its noise structure to optimize decision behavior. Third, it shows that the noise structure can remain unchanged irrespective of the amount of resources available to guide behavior (i.e., the noise distribution $f_\eta$ does not depend on $n$, *Equation 6*). Please note however, that this minimalistic implementation does not directly imply that the samples in our algorithmic formulation are necessarily drawn in this way. We believe that this implementation provides a simple demonstration of the consequences of limited resources in systems that encode information based on discrete stochastic events (*Sharpee, 2017*). Interestingly, it has been shown that this minimalistic formulation can be extended to more realistic population coding specifications (*Nikitin et al., 2009*).

## Efficient coding and the relation between environmental priors and discrimination

The results presented above imply that this encoding framework imposes limitations on the ability of capacity-limited systems to discriminate between different values of the encoded variables. Moreover, we have shown that error minimization in discrimination tasks implies a particular shape of the prior distribution of the encoder (*Equation 5*) that is exactly the prior density that maximizes mutual information between the input $v$ and the encoded noisy readings $k$ (*Equation 2*, *Figure 2a* right panel). Does this imply a relation between prior and discriminability over the space of the encoded

variable? Intuitively, following the efficient coding hypothesis, the relation should be that lower discrimination thresholds should occur for ranges of stimuli that occur more frequently in the environment or context.

Recently, it was shown that using an efficiency principle for encoding sensory variables (e.g., with a heterogeneous population of noisy neurons [*Ganguli and Simoncelli, 2016*]) it is possible to obtain an explicit relationship between the statistical properties of the environment and perceptual discriminability (*Ganguli and Simoncelli, 2016*). The theoretical relation states that discriminability thresholds $d$ should be inversely proportional to the density of the prior distribution $f(v)$. Here, we investigated whether this particular relation also emerges in the efficient coding scheme that we propose in this study.

Remarkably, we obtain the following relation between discriminability thresholds, prior distribution of input variables, and the number of limited samples $n$ (Appendix 4):

$$d = \frac{1}{\sqrt{n}\pi f(v)}$$
$$\propto \frac{1}{f(v)} \tag{7}$$

Interestingly, this relationship between prior distribution and discriminability thresholds holds empirically across several sensory modalities (Appendix 4), thus once again demonstrating that the efficient coding framework that we propose here seems to incorporate the right kind of constraints to explain observed perceptual phenomena as consequences of optimal allocation of finite capacity for internal representations.

## Maximizing the expected size of the selected option (fitness maximization)

Until now, we have studied the case when the goal of the organism is to minimize the number of mistakes in discrimination tasks. However, it is important to consider the case when the goal of the organism is to maximize fitness or expected reward (*Pirrone et al., 2014*). For example, when spending the day foraging fruit, one must make successive decisions about which tree has more fruits. Fitness depends on the number of fruit collected which is not a linear function of the number of accurate decisions, as each choice yields a different amount of fruit.

Therefore, in the case of reward maximization, we are interested in minimizing reward loss which is given by the following expression

$$\mathrm{E}[v(\text{chosen})] = \int \int f(v_1, v_2)[P_1(\theta(v_1), \theta(v_2))v_1 + P_2(\theta(v_1), \theta(v_2))v_2] \, dv_1 dv_2, \tag{8}$$

where $P_i(\theta(v_1), \theta(v_2))$ is the probability of choosing option $i$ when the input values are $v_1$ and $v_2$. Thus, the goal is to find the encoding rule $\theta(v)$ which guarantees that the amount of reward loss is as small as possible given our proposed coding framework.

Here we show that the optimal encoding rule $\theta(v)$ that guarantees maximization of expected value is given by

$$\theta(v) = \sin\left[\frac{\pi}{2} \cdot c \int_{-\infty}^{v} f(\tilde{v})^{2/3} d\tilde{v}\right]^2, \tag{9}$$

where $c$ is a normalizing constant which guarantees that the expression within the integral is a probability density function (Appendix 5). The first observation based on this result is that the encoding rule for maximizing fitness is different from the encoding rule that maximizes accuracy (compare *Equations 3 and 9*), which leads to a slight loss of information transmission (*Figure 2c*). Additionally, one can also obtain discriminability threshold predictions for this new encoding rule. Assuming a right-skewed prior distribution, which is often the case for various natural priors in the environment (e.g., like the one shown in *Figure 2a*), we find that discriminability for small input values is lower for reward maximization compared to perceptual maximization, however this pattern inverts for higher values (*Figure 2d*). In other words, when we intend to maximize reward (given the shape of our assumed prior, *Figure 2a*), the agent should allocate more resources to higher values (compared to the perceptual case), however without completely giving up sensitivity for lower values, as these values are still encountered more often.

## Efficient sampling with costs on acquiring prior knowledge

In the previous section, we obtained analytical solutions that approximately characterize the optimal $\theta(v)$ in the limit as $n$ is made sufficiently large. Note however that we are always assuming that is finite, and that this constrains the accuracy of the decision maker's judgments, while $m$ is instead unbounded and hence no constraint.

The nature of the optimal function $\theta(v_i; \tilde{v}_1, \ldots, \tilde{v}_m)$ is different, however, when $m$ is small. We argue that this scenario is particularly relevant when full knowledge of the prior is not warranted given the costs vs benefits of learning, for instance, when the system expects contextual changes to occur often. In this case, as we will formally elaborate below, it ceases to be efficient for $\theta$ to vary only gradually as a function of $v_i$, rather than moving abruptly from values near zero to values near one (Appendix 6). In the large-$m$ limiting case, the distributions of sample values $(\tilde{v}_1, \ldots, \tilde{v}_m)$ used by the different processing units will be nearly the same for each unit (approximating the current true distribution $f(v)$). Then if $\theta$ were to take only the values zero and one for different values of its arguments, the $n$ units would simply produce $n$ copies of the same output (either zero or one) for any given stimulus $v_i$ and distribution $f(v)$. Hence only a very coarse degree of differentiation among different stimulus magnitudes would be possible. Having $\theta$ vary more gradually over the range of values of $v_i$ in the support of $f(v)$ instead makes the representation more informative. But when $m$ is small (e.g., because of costs vs benefits of accurately representing the prior $f$), this kind of arbitrary randomization in the output of individual processing units is no longer essential. There will already be considerable variation in the outputs of the different units, even when the output of each unit is a deterministic function of $(v_i; \tilde{v}_1, \ldots, \tilde{v}_m)$, owing to the variability in the sample of prior observations that is used to assess the nature of the current environment. As we will show below, this variability will already serve to allow the collective output of the several units to differentiate between many gradations in the magnitude of $v_i$, rather than only being able to classify it as 'small' or 'large' (because either all units are in the 'low' or 'high' states).

### Robust optimality of decision by sampling

Because of the way in which sampling variability in the values $(\tilde{v}_1, \ldots, \tilde{v}_m)$ used to adapt each unit's encoding rule to the current context can substitute for the arbitrary randomization represented by the noise term $\eta_i$ (see *Figure 1*), a sharp reduction in the value of $m$ need not involve a great loss in performance relative to what would be possible (for the same limit on $n$) if $m$ were allowed to be unboundedly large (Appendix 7). As an example, consider the case in which $m = 1$, so that each unit $j$'s output state must depend only on the value of the current stimulus $v_i$ and one randomly selected draw $\tilde{v}_j$ from the prior distribution $f(v)$. A possible decision rule that is radically economical in this way is one that specifies that the unit will be in the 'high' state if and only if $v_i > \tilde{v}_j$. In this case, the internal representation of a stimulus $v_i$ will be given by the number $k_i$ out of $n$ independent draws from the contextual distribution $f(v)$ with the property that the contextual draw is smaller than $v_i$, as in the model of *decision by sampling* (DbS) (*Stewart et al., 2006*). However, it remains to be determined to what degree it might be beneficial for a system to adopt such coding strategy.

In any given environment (characterized by a particular contextual distribution $f(v)$), DbS will be equivalent to an encoding process with an architecture of the kind shown in *Figure 1*, but in which the distribution $f_\eta = f(v)$ (compare to the optimal noise distribution $f_\eta$ for the full prior adaptation case in *Equation 6*). This makes $\theta(v)$ vary endogenously depending on the contextual distribution $f(v)$. And indeed, the way that $\theta(v)$ varies with the contextual distribution under DbS is fairly similar to the way in which it would be optimal for it to vary in the absence of any cost of precisely learning and representing the contextual distribution. This result implies that $\theta(v)$ will be a monotonic transformation of a function that increases more steeply over those regions of the stimulus space where $f(v)$ is higher, regardless of the nature of the contextual distribution. We consider its performance in a given environment, from the standpoint of each of the possible performance criteria considered for the case of full prior adaptation (i.e., maximize accuracy or fitness), and show that it differs from the optimal encoding rules under any of those criteria (*Figure 2b–d*). In particular, here, we show that using the encoding rule employed in DbS results in considerable loss of information compared to the full-prior adaptation solutions (*Figure 2c*). An additional interesting observation is that for the strategy employed in DbS, the agent appears to be more sensitive for extreme input values, at least for a wide set of skewed distributions (e.g., for the prior distribution $f(v)$ in *Figure 2a*, the

discriminability thresholds are lower at the extremes of the support of $f(v)$). In other words, agents appear to be more sensitive to salience in the DbS rule. Despite these differences, here it is important to emphasize that in general for all optimization objectives, the encoding rules will be steeper for regions of the prior with higher density. However, mild changes in the steepness of the curves will be represented in significant discriminability differences between the different encoding rules across the support of the prior distribution (*Figure 2d*).

While the predictions of DbS are not exactly the same as those of efficient coding in the case of unbounded $m$, under any of the different objectives that we consider, our numerical results show that it can achieve performance nearly as high as that of the theoretically optimal encoding rule; hence radically reducing the value of $m$ does not have a large cost in terms of the accuracy of the decisions that can be made using such an internal representation (Appendix 7 and *Figure 2e*). Under the assumption that reducing either $m$ or $n$ would serve to economize on scarce cognitive resources, we formally prove that it might well be most efficient to use an algorithm with a very low value of $m$ (even $m = 1$, as assumed by DbS), while allowing $n$ to be much larger (Appendix 6, Appendix 7).

Crucially, here, it is essential to emphasize that the above-mentioned results are derived for the case of a particular finite number of processing units $n$ (and a corresponding finite total number of samples from the contextual distribution used to encode a given stimulus), and do not require that $n$ must be large (Appendix 6, Appendix 7).

## Testing theories of numerosity discrimination

Our goal now is to compare back-to-back the resource-limited coding frameworks elaborated above in a fundamental cognitive function for human behavior: numerosity perception. We designed a set of experiments that allowed us to test whether human participants would adapt their numerosity encoding system to maximize fitness or accuracy rates via full prior adaptation as usually assumed in optimal models, or whether humans employ a 'less optimal' but more efficient strategy such as DbS, or the more established logarithmic encoding model.

In Experiment 1, healthy volunteers (n = 7) took part in a two-alternative forced choice numerosity task in which each participant completed ~2400 trials across four consecutive days (Materials and methods). On each trial, they were simultaneously presented with two clouds of dots and asked which one contained more dots, and were given feedback on their reward and opportunity losses on each trial (*Figure 3a*). Participants were either rewarded for their accuracy (perceptual condition, where maximizing the amount of correct responses is the optimal strategy) or the number of dots they selected (value condition, where maximizing reward is the optimal strategy). Each condition was tested for two consecutive days with the starting condition randomized across participants. Crucially, we imposed a prior distribution $f(v)$ with a right-skewed quadratic shape (*Figure 3b*), whose parametrization allowed tractable analytical solutions of the encoding rules $\theta_A(v)$, $\theta_R(v)$ and $\theta_D(v)$, that correspond to the encoding rules for Accuracy maximization, Reward maximization, and DbS, respectively (*Figure 3e* and Materials and methods). Qualitative predictions of behavioral performance indicate that the accuracy-maximization model is the most accurate for trials with lower numerosities (the most frequent ones), whereas the reward-maximization model outperforms the others for trials with larger numerosities (trials where the difference in the number of dots in the clouds, and thus the potential reward, is the largest, *Figure 2d* and *Figure 3f*). In contrast, the DbS strategy presents markedly different performance predictions, in line with the discriminability predictions of our formal analyses (*Figure 2c,d*).

In our modelling specification, the choice structure is identical for the three different sampling models, differing only in the encoding rule $\theta(v)$ (Materials and methods). Therefore, answering the question of which encoding rule is the most favored for each participant can be parsimoniously addressed using a latent-mixture model, where each participant uses $\theta_A(v)$, $\theta_R(v)$ or $\theta_D(v)$ to guide their decisions (Materials and methods). Before fitting this model to the empirical data, we confirmed the validity of our model selection approach through a validation procedure using synthetic choice data (*Figure 3d*, *Figure 3—figure supplement 1*, and Materials and methods).

After we confirmed that we can reliably differentiate between our competing encoding rules, the latent-mixture model was initially fitted to each condition (perceptual or value) using a hierarchical Bayesian approach (Materials and methods). Surprisingly, we found that participants did not follow the accuracy or reward optimization strategy in the respective experimental condition, but favored the DbS strategy (proportion that DbS was deemed best in the perceptual $p_{\text{DbSfavored}} = 0.86$ and

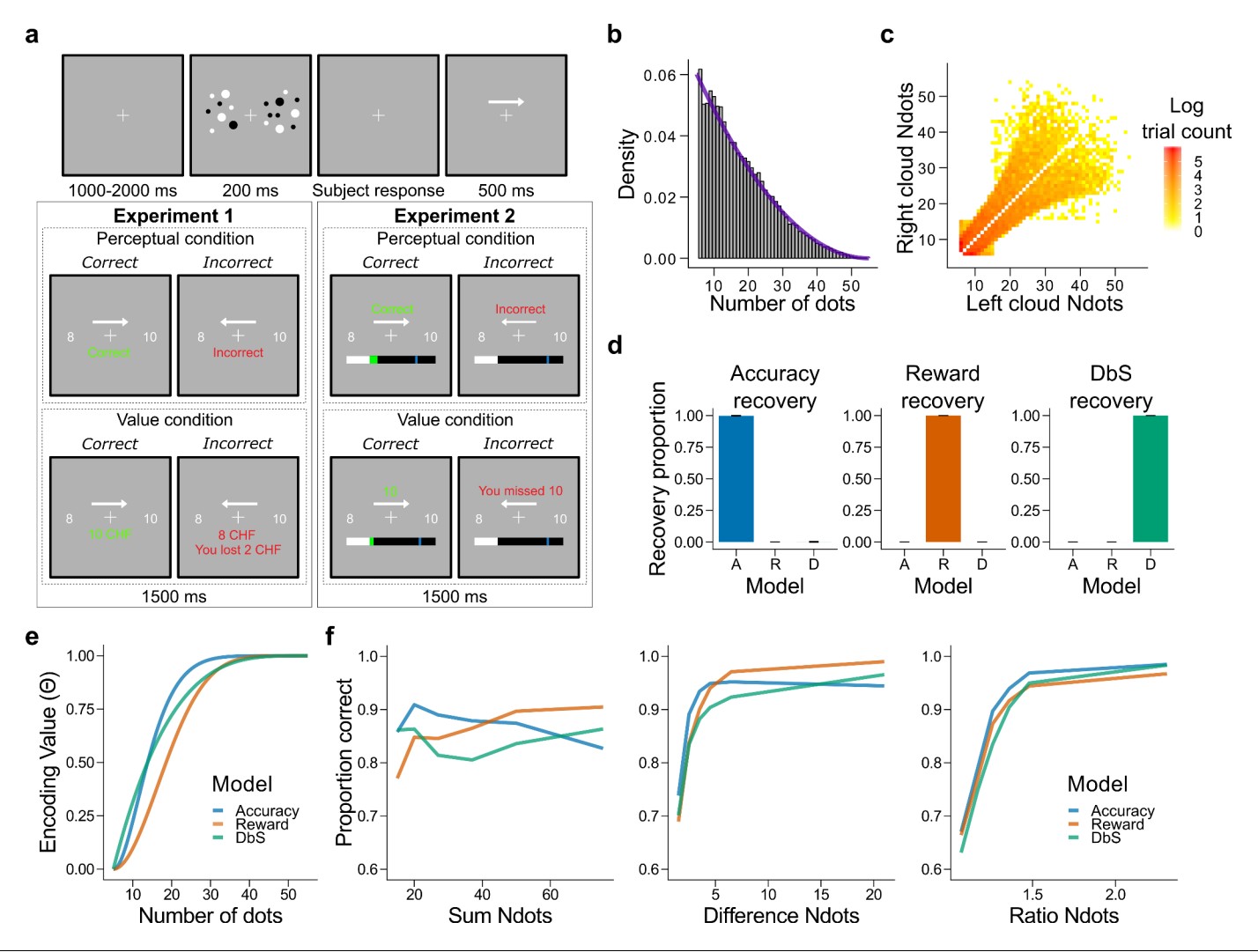

**Figure 3.** Experimental design, model simulations and recovery. (a) Schematic task design of Experiments 1 and 2. After a fixation period (1–2 s) participants were presented two clouds of dots (200 ms) and had to indicate which cloud contained the most dots. Participants were rewarded for being accurate (perceptual condition) or for the number of dots they selected (value condition) and were given feedback. In Experiment 2 participants collected on correctly answered trials a number of points equal to a fixed amount (perceptual condition) or a number equal to the dots in the cloud they selected (value condition) and had to reach a threshold of points on each run. (b) Empirical (grey bars) and theoretical (purple line) distribution of the number of dots in the clouds of dots presented across Experiments 1 and 2. (c) Distribution of the numerosity pairs selected per trial. (d) Synthetic data preserving the trial set statistics and number of trials per participant used in Experiment 1 was generated for each encoding rule (Accuracy (left), Reward (middle), and DbS (right)) and then the latent-mixture model was fitted to each generated dataset. The figures show that it is theoretically possible to recover each generated encoding rule. (e) Encoding function $\theta(v)$ for the different sampling strategies as a function of the input values $v$ (i. e., the number of dots). (f) Qualitative predictions of the three models (blue: Accuracy, red: Reward, green: Decision by Sampling) on trials from Experiment 1 with $n = 25$. Performance of each model as a function of the sum of the number of dots in both clouds (left), the absolute difference between the number of dots in both clouds (middle) and the ratio of the number of dots in the most numerous cloud over the less numerous cloud (right).

The online version of this article includes the following figure supplement(s) for figure 3:

**Figure supplement 1.** Model recovery for $\alpha$ fixed.

**Figure supplement 2.** Model recovery with both $\alpha$ and $n$ as free parameters.

**Figure supplement 3.** Discriminability differences between the different encoding rules.

value $p_{\mathrm{DbSfavored}} = 0.93$ conditions, *Figure 4*). Importantly, this population-level result also holds at the individual level: DbS was strongly favored in 6 out of 7 participants in the perceptual condition, and seven out of seven in the value condition (*Figure 4—figure supplement 1*). These results are not likely to be affected by changes in performance over time, as performance was stable across the four consecutive days (*Figure 4—figure supplement 2*). Additionally, we investigated whether biases induced by choice history effects may have influenced our results (*Abrahamyan et al., 2016*; *Keung et al., 2019*; *Talluri et al., 2018*). Therefore, we incorporated both choice- and correctness-dependence history biases in our models and fitted the models once again (Materials and methods). We found similar results to the history-free models ($p_{\mathrm{DbSfavored}} = 0.87$ in perceptual and $p_{\mathrm{DbSfavored}} = 0.93$ in value conditions, *Figure 4c*). At the individual level, DbS was again strongly favored in 6 out of 7 participants in the perceptual condition, and 7 out of 7 in the value condition (*Figure 4—figure supplement 1*).

In order to investigate further the robustness of this effect, we introduced a slight variation in the behavioral paradigm. In this new experiment (Experiment 2), participants were given points on each trial and had to reach a certain threshold in each run for it to be eligible for reward (*Figure 3a* and Materials and methods). This class of behavioral task is thought to be in some cases more ecologically valid than trial-independent choice paradigms (*Kolling et al., 2014*). In this new experiment, either a fixed amount of points for a correct trial was given (perceptual condition) or an amount equal to the number of dots in the chosen cloud if the response was correct (value condition). We recruited a new set of participants (n = 6), who were tested on these two conditions, each for two consecutive days with the starting condition randomized across participants (each participant completed $\sim 2,560$ trials). The quantitative results revealed once again that participants did not change their encoding strategy depending on the goals of the task, with DbS being strongly favored for both perceptual and value conditions ($p_{\mathrm{DbSfavored}} = 0.999$ and $p_{\mathrm{DbSfavored}} = 0.91$, respectively; *Figure 4a*), and these results were confirmed at the individual level where DbS was strongly favored in 6 out of 6 participants in both the perceptual and value conditions (*Figure 4—figure supplement 1*). Once again, we found that inclusion of choice history biases in this experiment did not significantly affect our results both at the population and individual levels. Population probability that DbS was deemed best in the perceptual ($p_{\mathrm{DbSfavored}} = 0.999$) and value ($p_{\mathrm{DbSfavored}} = 0.90$) conditions (*Figure 4—figure supplement 1*), and at the individual level DbS was strongly favored in 6 out of 6 participants in the perceptual condition and 5 of 6 in the value condition (*Figure 4—figure supplement 1*). Thus, Experiments 1 and 2 strongly suggest that our results are not driven by specific instructions or characteristics of the behavioral task.

As a further robustness check, for each participant we grouped the data in different ways across experiments (Experiments 1 and 2) and experimental conditions (perceptual or value) and investigated which sampling model was favored. We found that irrespective of how the data was grouped, DbS was the model that was clearly deemed best at the population (*Figure 4*) and individual level (*Figure 4—figure supplement 3*). Additionally, we investigated whether these quantitative results specifically depended on our choice of using a latent-mixture model. Therefore, we also fitted each model independently and compared the quality of the model fits based on out-of-sample cross-validation metrics (Materials and methods). Once again, we found that the DbS model was favored independently of experiment and conditions (*Figure 4*).

One possible reason why the two experimental conditions did not lead to differences could be that, after doing one condition for two days, the participants did not adapt as easily to the new incentive rule. However, note that as the participants did not know of the second condition before carrying it out, they could not adopt a compromise between the two behavioral objectives. Nevertheless, we fitted the latent-mixture model only to the first condition that was carried out by each participant. We found once again that DbS was the best model explaining the data, irrespective of condition and experimental paradigm (*Figure 4—figure supplement 7*). Therefore, the fact that DbS is favored in the results is not an artifact of carrying out two different conditions in the same participants.

We also investigated whether the DbS model makes more accurate predictions than the widely used logarithmic model of numerosity discrimination tasks (*Dehaene, 2003*). We found that DbS still made better out-of-sample predictions than the log-model (*Figure 4b*, *Figure 5f,g*). Moreover, these results continued to hold after taking into account possible choice history biases (*Figure 4—figure supplement 4*). In addition to these quantitative results, qualitatively we also found that

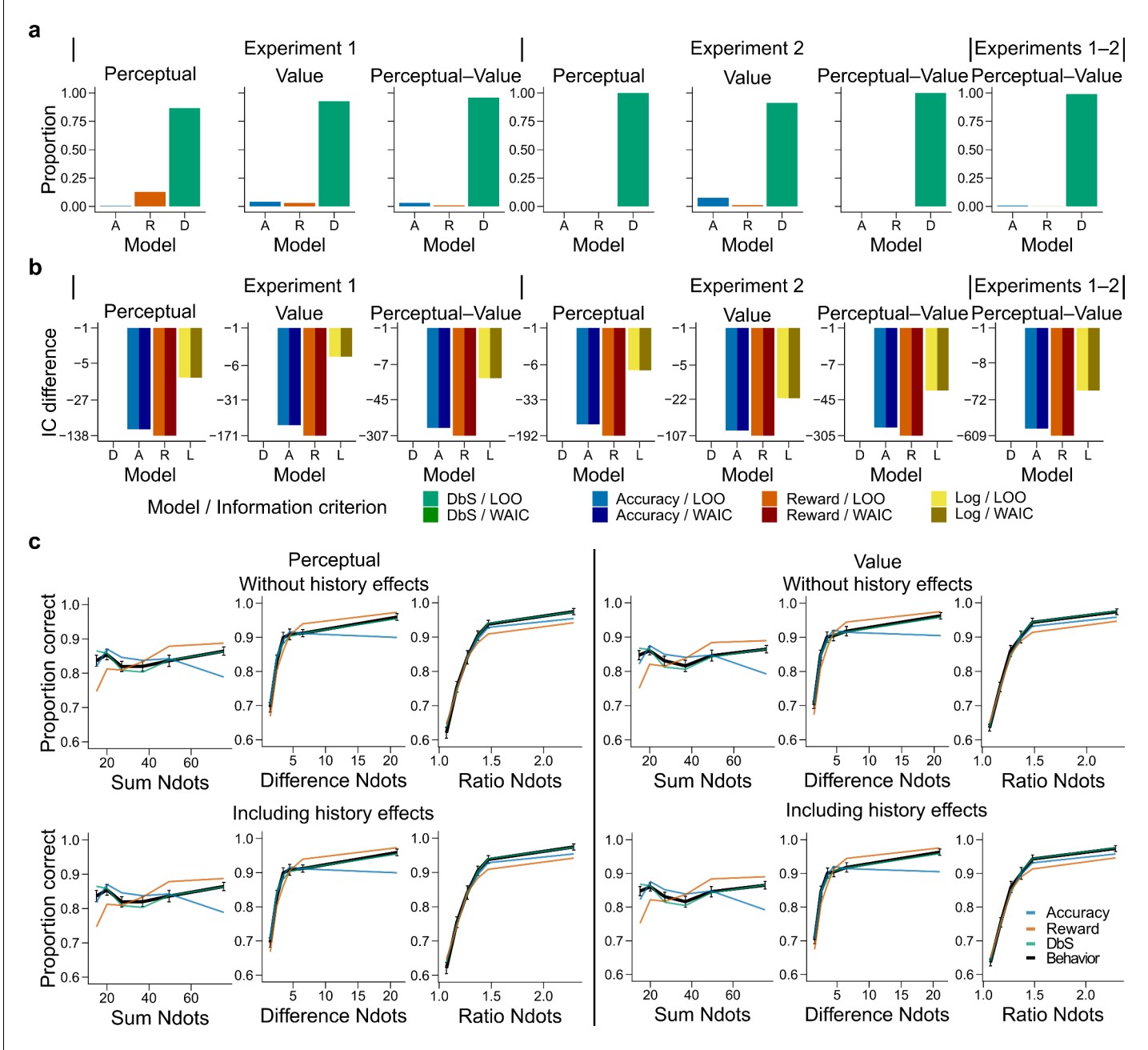

**Figure 4.** Behavioral results. (**a**) Bars represent proportion of times an encoding rule (Accuracy [A, blue], Reward [R, red], DbS [D, green]) was selected by the Bayesian latent-mixture model based on the posterior estimates across participants. Each panel shows the data grouped for each and across experiments and experimental conditions (see titles on top of each panel). The results show that DbS was clearly the favored encoding rule. The latent vector $\pi$ posterior estimates are presented in *Figure 4—figure supplement 4*. (**b**) Difference in LOO and WAIC between the best model (DbS (D) in all cases) and the competing models: Accuracy (A), Reward (R) and Logarithmic (L) models. Each panel shows the data grouped for each and across experimental conditions and experiments (see titles on top of each panel). (**c**) Behavioral data (black, error bars represent SEM across participants) and model predictions based on fits to the empirical data. Data and model predictions are presented for both the perceptual (left panels) or value (right panels) conditions, and excluding (top panels) or including (bottom panels) choice history effects. Performance of data model predictions is presented as function of the sum of the number of dots in both clouds (left), the absolute difference between the number of dots in both clouds (middle) and the ratio of the number of dots in the most numerous cloud over the less numerous cloud (right). Results reveal a remarkable overlap of the behavioral data and predictions by DbS, thus confirming the quantitative results presented in panels a and b.

The online version of this article includes the following figure supplement(s) for figure 4:

**Figure supplement 1.** Latent mixture model fits for each participant.

**Figure supplement 2.** Performance across time.

*Figure 4 continued on next page*

*Figure 4 continued*

**Figure supplement 3.** Individual level fit of the latent mixture model combining data across experiments and experimental conditions.
**Figure supplement 4.** Model comparison based on leave-one-out cross-validation metrics.
**Figure supplement 5.** Reaction times are similar in the perceptual and value conditions.
**Figure supplement 6.** Behavior and model predictions as a function of sum and difference in dots.
**Figure supplement 7.** Model fit for the first experimental condition of each participant.
**Figure supplement 8.** Latent vector $\pi$ posterior estimates.

behavior closely matched the predictions of the DbS model remarkably well (*Figure 4c*), based on virtually only one free parameter, namely, the number of samples (resources) $n$. Together, these results provide compelling evidence that DbS is the most likely resource-constrained sampling strategy used by participants in numerosity discrimination tasks.

Recent studies have also investigated behavior in tasks where perceptual and preferential decisions have been investigated in paradigms with identical visual stimuli (*Dutilh and Rieskamp, 2016*; *Polanía et al., 2014*; *Grueschow et al., 2015*). In these tasks, investigators have reported differences in behavior, in particular in the reaction times of the responses, possibly reflecting differences in behavioral strategies between perceptual and value-based decisions. Therefore, we investigated whether this was the case also in our data. We found that reaction times did not differ between experimental conditions for any of the different performance assessments considered here (*Figure 4—figure supplement 5*). This further supports the idea that participants were in fact using the same sampling mechanism irrespective of behavioral goals.

Here it is important to emphasize that all sampling models and the logarithmic model of numerosity have the same degrees of freedom (performance is determined by $n$ in the sampling models and Weber's fraction $\sigma$ in the log model, Materials and methods). Therefore, qualitative and quantitative differences favoring the DbS model cannot be explained by differences in model complexity. It could also be argued that normal approximation of the binomial distributions in the sampling decision models only holds for large enough $n$. However, we find evidence that the large-$n$ optimal solutions are also nearly optimal for low $n$ values (Appendix 7). Estimates of $n$ in our data are in general $n \approx 21$ (*Table 1*) and we find that the large-$n$ rule is nearly optimal already for $n = 15$ (Appendix 7). Therefore the asymptotic approximations should not greatly affect the conclusions of our work.

## Dynamics of adaptation

Up to now, fits and comparison across models have been done under the assumption that the participants learned the prior distribution $f(v)$ imposed in our task. If participants are employing DbS, it is important to understand the dynamical nature of adaptation in our task. Note that the shape of the prior distribution is determined by the parameter $\alpha$ (*Figure 5b*, *Equation 10* in Materials and methods). First, we made sure based on model recovery analyses that the DbS model could jointly and accurately recover both the shape parameter $\alpha$ and the resource parameter $n$ based on synthetic data (*Figure 3—figure supplement 2*). Then we fitted this model to the empirical data and found that the recovered value of the shape parameter $\alpha$ closely followed the value of the empirical prior with a slight underestimation (*Figure 5a*). Next, we investigated the dynamics of prior adaptation. To this end, we ran a new experiment (Experiment 3, n = 7 new participants) in which we set the shape parameter of the prior to a lower value compared to Experiments 1–2 (*Figure 5b*, Materials and methods). We investigated the change of $\alpha$ over time by allowing this parameter to change with trial experience (*Equation 18*, Materials and methods) and compared the evolution of $\alpha$ for Experiments 1 and 2 (empirical $\alpha = 2$) with Experiment 3 (empirical $\alpha = 1$, *Figure 5b*). If participants show prior adaptation in our numerosity discrimination task, we hypothesized that the asymptotic value of $\alpha$ should be higher for Experiments 1–2 than for Experiment 3. First, we found that for Experiments 1–2, the value of $\alpha$ quickly reached an asymptotic value close to the target value (*Figure 5c*). On the other hand, for Experiment 3 the value of $\alpha$ continued to decrease during the experimental session, but slowly approaching its target value. This seemingly slower adaptation to the shape of the prior in Experiment 3 might be explained by the following observation. The prior parametrized with $\alpha = 1$ in Experiment 3 is further away from an agent hypothesized to have a natural numerosity discrimination based on a log scale ($\alpha = 2.58$, Materials and methods), which is closer in value to the shape of the prior in Experiments 1 and 2 ($\alpha = 2$).

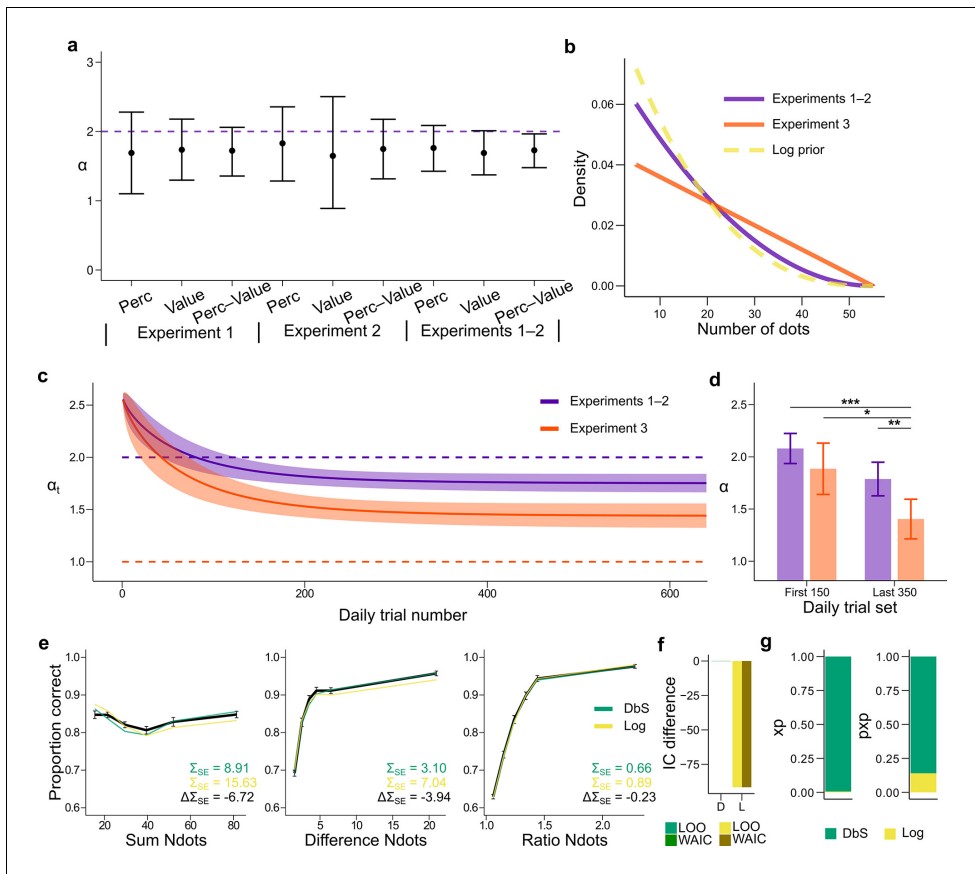

**Figure 5.** Prior adaptation analyses. (**a**) Estimation of the shape parameter $\alpha$ for the DbS model by grouping the data for each and across experimental conditions and experiments. Error bars represent the 95% highest density interval of the posterior estimate of $\alpha$ at the population level. The dashed line shows the theoretical value of $\alpha$. (**b**) Theoretical prior distribution $f(v)$ in Experiments 1 and 2 ($\alpha = 2$, purple) and 3 ($\alpha = 1$, orange). The dashed line represents the value of $\alpha$ of our prior parametrization that approximates the DbS and log discriminability models. (**c**) Posterior estimation of $\alpha_t$ (***Equation 18***) as a function of the number of trials $t$ in each daily session for Experiments 1 and 2 (purple) and Experiment 3 (orange). The results reveal that, as expected, $\alpha_t$ reaches a lower asymptotic value $\delta$. Error bars represent ± SD of 3000 simulated $\alpha_t$ values drawn from the posterior estimates of the HBM (see Materials and methods). (**d**) Model fit to the first 150 and last 350 trials of each daily session. The $\alpha$ parameter was allowed to vary between the first and last sets of daily trials and between Experiments 1–2 and Experiment 3. In Experiment 3, $\alpha$ is lower in the last set of trials compared to the first set of trials ($P_{\mathrm{MCMC}} = 0.013$). In addition, $\alpha$ for the last trials is lower for Experiment 3 than for Experiments 1–2 ($P_{\mathrm{MCMC}} = 0.006$). This confirms that the results presented in panel c are not artifacts of the adaptation parametrization assumed for $\alpha$. Error bars represent ± SD of the posterior chains of the corresponding parameter. (*P<0.05, **P<0.01, and ***P<0.001). (**e**) Behavioral data (black) and model fit predictions of the DbS (green) and Log (yellow) models. Performance of each model as a function of the sum of the number of dots in both clouds (left), the absolute difference between the number of dots in both clouds (middle) and the ratio of the number of dots in the most numerous cloud over the less numerous cloud (right). Error bars represent SEM (**f**) Difference in LOO and WAIC between the best fitting DbS (D) and logarithmic encoding (Log) model. (**g**) Population exceedance probabilities (xp, left) and protected exceedance probabilities (pxp, right) for DbS (green) vs Log (yellow) of a Bayesian model selection analysis (***Stephan et al., 2009***): $\mathrm{xp_{DbS}} = 0.99$, $\mathrm{pxp_{DbS}} = 0.87$. These results provide a clear indication that the adaptive DbS explains the data better than the Log model.

The online version of this article includes the following figure supplement(s) for figure 5:

**Figure supplement 1.** Performance across trial experience.

**Figure supplement 2.** Quantitative and dynamical analysis of adaptation over time.

**Figure supplement 3.** Model fits for the beginning and end of each session without parametric assumptions.

**Table 1.** Resource parameter $n$ fits.

Fits of the resource parameter for the Accuracy, Reward and Decision by Sampling (DbS) models including data across experiments and conditions (perceptual (P) or value (V)) either including or ignoring choice history effects. The values represent the mean ± SD of the posterior distributions at the population level for parameter $n$. Note that Reward and in particular the DbS encoding models require a higher number of resources than the Accuracy model, which is coherent with the fact that the Accuracy model allocates its resources to maximize efficiency, therefore reducing the number of resources needed to reach a given accuracy. DbS has the highest values of $n$ because it is the most inefficient model.

| Experiment | Condition | History effects | Model $n_{Accuracy}$ | $n_{Reward}$ | $n_{DbS}$ |
|---|---|---|---|---|---|
| 1 | V | not included | 15.24 ± 3.09 | 17.54 ± 3.98 | 24.40 ± 5.16 |
| 2 | V | not included | 22.48 ± 2.43 | 27.58 ± 3.81 | 35.40 ± 3.44 |
| 1 | P | not included | 15.19 ± 3.99 | 17.84 ± 4.85 | 24.64 ± 6.59 |
| 2 | P | not included | 20.99 ± 1.59 | 24.22 ± 1.93 | 33.54 ± 2.45 |
| 1 | P/V | not included | 15.33 ± 3.41 | 17.25 ± 4.45 | 24.15 ± 5.75 |
| 2 | P/V | not included | 21.30 ± 0.96 | 25.27 ± 1.99 | 33.90 ± 1.51 |
| 1/2 | V | not included | 18.56 ± 2.04 | 22.05 ± 2.73 | 29.52 ± 3.25 |
| 1/2 | P | not included | 17.91 ± 2.09 | 20.66 ± 2.59 | 28.62 ± 3.51 |
| 1/2 | P/V | not included | 17.93 ± 1.87 | 21.03 ± 2.46 | 28.58 ± 3.04 |
| 1 | V | included | 15.50 ± 3.13 | 17.50 ± 3.91 | 24.68 ± 5.08 |
| 2 | V | included | 22.92 ± 2.37 | 28.07 ± 3.73 | 36.18 ± 2.91 |
| 1 | P | included | 15.41 ± 3.81 | 17.96 ± 4.88 | 24.70 ± 6.62 |
| 2 | P | included | 21.57 ± 1.71 | 24.88 ± 2.17 | 34.37 ± 2.93 |
| 1 | P/V | included | 15.16 ± 3.55 | 17.43 ± 4.39 | 24.30 ± 5.94 |
| 2 | P/V | included | 21.80 ± 0.92 | 25.81 ± 1.86 | 34.60 ± 1.40 |
| 1/2 | V | included | 18.86 ± 2.07 | 22.48 ± 2.75 | 29.85 ± 3.17 |
| 1/2 | P | included | 18.15 ± 2.17 | 21.11 ± 2.72 | 29.01 ± 3.47 |
| 1/2 | P/V | included | 18.22 ± 1.93 | 21.34 ± 2.50 | 29.12 ± 3.12 |

Irrespective of these considerations, the key result to confirm our adaptation hypothesis is that the asymptotic value of $\alpha$ is lower for Experiment 3 compared to Experiments 1 and 2 ($P_{\mathrm{MCMC}} = 0.006$).

In order to make sure that this result was not an artifact of the parametric form of adaptation assumed here (*Equation 18*, Materials and methods), we fitted the DbS model to trials at the beginning and end of each experimental session allowing $\alpha$ to be a free but fixed parameter in each set of trials. The results of these new analyses are virtually identical to the results obtained with the parametric form, in which $\alpha$ is smaller at the end of Experiment 3 sessions relative to beginning of Experiments 1 and 2 ($P_{\mathrm{MCMC}} = 0.0003$), beginning of Experiments 3 ($P_{\mathrm{MCMC}} = 0.013$) and end of Experiments 1 and 2 ($P_{\mathrm{MCMC}} = 0.006$, *Figure 5d*). In this model, we did not allow $n$ to freely change for each condition, and therefore a concern might be that the results might be an artifact of changes in $n$, which could for example change with the engagement of the participants across the session. Given that we already demonstrated that both parameters $n$ and $\alpha$ are identifiable, we fitted the same model as in *Figure 5d*, however this time we allowed $n$ to be free parameter alongside $\alpha$. We found that the results obtained in *Figure 5d* remained virtually unchanged (*Figure 5—figure supplement 3*), in addition to the result that the resource parameter $n$ remained virtually identical across the session (*Figure 5—figure supplement 3*).

We further investigated evidence for adaptation using an alternative quantitative approach. First, we performed out-of-sample model comparisons based on the following models: (i) the adaptive-$\alpha$ model, (ii) free-$\alpha$ model with $\alpha$ free but non-adapting over time, and (iii) fixed-$\alpha$ model with $\alpha = 2$. The results of the out-of-sample predictions revealed that the best model was the free-$\alpha$ model,

followed closely by the adaptive-$\alpha$ model ($\Delta\text{LOO} = 1.8$) and then by fixed-$\alpha$ model ($\Delta\text{LOO} = 32.6$). However, we did not interpret the apparent small difference between the adaptive-$\alpha$ and the free-$\alpha$ models as evidence for lack of adaptation, given that the more complex adaptive-$\alpha$ model will be strongly penalized after adaptation is stable. That is, if adaptation is occurring, then the adaptive-$\alpha$ only provides a better fit for the trials corresponding to the adaptation period. After adaptation, the adaptive-$\alpha$ should provide a similar fit than the free-$\alpha$ model, however with a larger complexity that will be penalized by model comparison metrics. Therefore, to investigate the presence of adaptation, we took a closer quantitative look at the evolution of the fits across trial experience. We computed the average trial-wise predicted Log-Likelihood (by sampling from the hierarchical Bayesian model) and compared the differences of this metric between the competing models and the adaptive model. We hypothesized that if adaptation is taking place, the adaptive-$\alpha$ model would have an advantage relative to the free-$\alpha$ model at the beginning of the session, with these differences vanishing toward the end. On the other hand, the fixed-$\alpha$ should roughly match the adaptive-$\alpha$ model at the beginning and then become worse over time, but these differences should stabilize after the end of the adaptation period. The results of these analyses support our hypotheses (*Figure 5—figure supplement 2*), thus providing further evidence of adaptation, highlighting the fact that the DbS model can parsimoniously capture adaptation to contextual changes in a continuous and dynamical manner. Furthermore, we found that the DbS model again provides more accurate qualitative and quantitative out-of-sample predictions than the log model (*Figure 5e,f*).

## Discussion

The brain is a metabolically expensive inference machine (*Hawkes et al., 1998*; *Navarrete et al., 2011*; *Stone, 2018*). Therefore, it has been suggested that evolutionary pressure has driven it to make productive use of its limited resources by exploiting statistical regularities (*Attneave, 1954*; *Barlow, 1961*; *Laughlin, 1981*). Here, we incorporate this important — often ignored — aspect in models of behavior by introducing a general framework of decision-making under the constraints that the system: (i) encodes information based on binary codes, (ii) has limited number of samples available to encode information, and (iii) considers the costs of contextual adaptation.

Under the assumption that the organism has fully adapted to the statistics in a given context, we show that the encoding rule that maximizes mutual information is the same rule that maximizes decision accuracy in two-alternative decision tasks. However, note that there is nothing privileged about maximizing mutual information, as it does not mean that the goals of the organism are necessarily achieved (*Park and Pillow, 2017*; *Salinas, 2006*). In fact, we show that if the goal of the organism is instead to maximize the expected value of the chosen options, the system should not rely on maximizing information transmission and must give up a small fraction of precision in information coding. Here, we derived analytical solution for each of these optimization objective criteria, emphasizing that these analytical solutions were derived for the large-$n$ limiting case. However, we have provided evidence that these solutions continue to be more efficient relative to DbS for small values of $n$, and more importantly, they remain nearly optimal even at relatively low values of $n$, in the range of values that might be relevant to explain human experimental data (Appendix 7).

Another key implication of our results is that we provide an alternative explanation to the usual conception of noise as the main cause of behavioral performance degradation, where noise is usually artificially added to models of decision behavior to generate the desired variability (*Ratcliff and Rouder, 1998*; *Wang, 2002*). On the contrary, our work makes it formally explicit why a system that evolved to encode information based on binary codes must be necessarily noisy, also revealing how the system could take advantage of its unavoidable noisy properties (*Faisal et al., 2008*) to optimize decision behavior (*Tsetsos et al., 2016*). Here, it is important to highlight that this conclusion is drawn from a purely homogeneous neural circuit, in other words, a circuit in which all neurons have the same properties (in our case, the same activation thresholds). This is not what is typically observed, as neural circuits are typically very heterogeneous. However, in the neural circuit that we consider here, it could mean that the firing thresholds can vary across neurons (*Orbán et al., 2016*), which could be used by the system to optimize the required variability of binary neural codes. Interestingly, it has been shown in recent work that stochastic discrete events also serve to optimize information transmission in neural population coding (*Ashida and Kubo, 2010*; *Nikitin et al., 2009*; *Schmerl and McDonnell, 2013*). Crucially, in our work we provide a direct link of the necessity of

noise for systems that aim at optimizing decision behavior under our encoding and limited-capacity assumptions, which can be seen as algorithmic specifications of the more realistic population coding specifications mentioned above (*Nikitin et al., 2009*). We argue that our results may provide a formal intuition for the apparent necessity of noise for improving training and learning performance in artificial neural networks (*Dapello et al., 2020*; *Findling and Wyart, 2020*), and we speculate that an implementation of 'the right' noise distribution for a given environmental statistic could be seen as a potential mechanism to improve performance in capacity-limited agents generally speaking (*Garrett et al., 2011*). We acknowledge that based on the results of our work, we cannot confirm whether this is the case for higher order neural circuits, however, we leave it as an interesting theoretical formulation, which could be addressed in future work.

Interestingly, our results could provide an alternative explanation of the recent controversial finding that dynamics of a large proportion of LIP neurons likely reflect binary (discrete) coding states to guide decision behavior (*Latimer et al., 2015*; *Zoltowski et al., 2019*). Based on this potential link between their work and ours, our theoretical framework generates testable predictions that could be investigated in future neurophysiological work. For instance, noise distribution in neural circuits should dynamically adapt according to the prior distribution of inputs and goals of the organism. Consequently, the rate of 'step-like' coding in single neurons should also be dynamically adjusted (perhaps optimally) to statistical regularities and behavioral goals.

Our results are closely related to Decision by Sampling (DbS), which is an influential account of decision behavior derived from principles of retrieval and memory comparison by taking into account the regularities of the environment, and also encodes information based on binary codes (*Stewart et al., 2006*). We show that DbS represents a special case of our more general efficient sampling framework, that uses a rule that is similar to (though not exactly like) the optimal encoding rule that assumes full (or costless) adaptation to the prior statistics of the environment. In particular, we show that DbS might well be the most efficient sampling algorithm, given that a reduction in the full representation of the prior distribution might not come at a great loss in performance. Interestingly, our experimental results (discussed in more detail below) also provide support for the hypothesis that numerosity perception is efficient in this particular way. Crucially, DbS automatically adjusts the encoding in response to changes in the frequency distribution from which exemplars are drawn in approximately the right way, while providing a simple answer to the question of how such adaptation of the encoding rule to a changing frequency distribution occurs, at a relatively low cost.

On a related line of work, *Bhui and Gershman, 2018* develop a similar, but different specification of DbS, in which they also consider only a finite number of samples that can be drawn from the prior distribution to generate a percept, and ask what kind of algorithm would be required to improve coding efficiency. However, their implementation differs from ours in various important ways (see Appendix 8 for a detailed discussion). One of the main distinctions is that they consider the case in which only a finite number of samples can be drawn from the prior and show that a variant of DbS with kernel-smoothing is superior to its standard version. However, a key difference to our implementation is that they allow the kernel-smoothed quantity (computed by comparing the input $v$ with a sample $\tilde{v}$ from the prior distribution) to vary continuously between 0 and 1, rather than having to be either 0 or 1 as in our implementation (*Figure 1*). Thus, they show that coding efficiency can be improved by allowing a more flexible implementation of the coding scheme for the case when the agent is allowed to draw few samples from the prior distribution (Appendix 8). On the other hand, we restrict our framework to a coding scheme that is only allowed to encode information based on zeros or ones, where we show that coding efficiency can be improved relative to DbS only under a more complete knowledge of the prior distribution, where the optimal solutions can be formally derived in the large-$n$ limit. Nevertheless, we have shown that even under the operation of few sampling units, the optimal rules will be still superior to the standard DbS (if the agent has fully adapted to the statistics of the environment in a given context), even when a few number of processing units are available to generate decision relevant percepts.

We tested these resource-limited coding frameworks in non-symbolic numerosity discrimination, a fundamental cognitive function for behavior in humans and other animals, which may have emerged during evolution to support fitness maximization (*Nieder, 2020*). Here, we find that the way in which the precision of numerosity discrimination varies with the size of the numbers being compared is consistent with the hypothesis that the internal representations on the basis of which comparisons are made are sample-based. In particular, we find that the encoding rule varies

depending on the frequency distribution of values encountered in a given environment, and that this adaptation occurs fairly quickly once the frequency distribution changes.

This adaptive character of the encoding rule differs, for example, from the common hypothesis of a logarithmic encoding rule (independent of context), which we show fits our data less well. Nonetheless, we can reject the hypothesis of full optimality of the encoding rule for each distribution of values used in our experiments, even after participants have had extensive experience with a given distribution. Thus, a possible explanation of why DbS is the favored model in our numerosity task is that accuracy and reward maximization requires optimal adaptation of the noise distribution based on our imposed prior, requiring complex neuroplastic changes to be implemented, which are in turn metabolically costly (*Buchanan et al., 2013*). Relying on samples from memory might be less metabolically costly as these systems are plastic in short time scales, and therefore a relatively simpler heuristic to implement allowing more efficient adaptation. Here, it is important to emphasize, as it has been discussed in the past (*Tajima et al., 2016*; *Polanía et al., 2015*), that for decision-making systems beyond the perceptual domain, the identity of the samples is unclear. We hypothesize, that information samples derive from the interaction of memory on current sensory evidence depending on the retrieval of relevant samples to make predictions about the outcome of each option for a given behavioral goal (therefore also depending on the encoding rule that optimizes a given behavioral goal).

Interestingly, it was recently shown that in a reward learning task, a model that estimates values based on memory samples from recent past experiences can explain the data better than canonical incremental learning models (*Bornstein et al., 2017*). Based on their and our findings, we conclude that sampling from memory is an efficient mechanism for guiding choice behavior, as it allows quick learning and generalization of environmental contexts based on recent experience without significantly sacrificing behavioral performance. However, it should be noted that relying on such mechanisms alone might be suboptimal from a performance- and goal-based point of view, where neural calibration of optimal strategies may require extensive experience, possibly via direct interactions between sensory, memory and reward systems (*Gluth et al., 2015*; *Saleem et al., 2018*).

Taken together, our findings emphasize the need of studying optimal models, which serve as anchors to understand the brain's computational goals without ignoring the fact that biological systems are limited in their capacity to process information. We addressed this by proposing a computational problem, elaborating an algorithmic solution, and proposing a minimalistic implementational architecture that solves the resource-constrained problem. This is essential, as it helps to establish frameworks that allow comparing behavior not only across different tasks and goals, but also across different levels of description, for instance, from single cell operation to observed behavior (*Marr, 1982*). We argue that this approach is fundamental to provide benchmarks for human performance that can lead to the discovery of alternative heuristics (*Qamar et al., 2013*; *Gardner, 2019*) that could appear to be in principle suboptimal, but that might be in turn the optimal strategy to implement if one considers cognitive limitations and costs of optimal adaptation. We conclude that the understanding of brain function and behavior under a principled research agenda, which takes into account decision mechanisms that are biologically feasible, will be essential to accelerate the elucidation of the mechanisms underlying human cognition.

## Materials and methods

### Participants

The study tested young healthy volunteers with normal or corrected-to-normal vision (total n = 20, age 19–36 years, nine females: n = 7 in Experiment 1, two females; n = 6 new participants in Experiment 2, three females; n = 7 new participants in Experiment 3, four females). Participants were randomly assigned to each experiment and no participant was excluded from the analyses. Participants were instructed about all aspects of the experiment and gave written informed consent. None of the participants suffered from any neurological or psychological disorder or took medication that interfered with participation in our study. Participants received monetary compensation for their participation in the experiment partially related to behavioral performance (see below). The experiments conformed to the Declaration of Helsinki and the experimental protocol was approved by the Ethics Committee of the Canton of Zurich (BASEC: 2018–00659).

## Experiment 1

Participants (n = 7) carried out a numerosity discrimination task for four consecutive days for approximately one hour per day. Each daily session consisted of a training run followed by 8 runs of 75 trials each. Thus, each participant completed ~2400 trials across the four days of experiment.

After a fixation period (1–1.5 s jittered), two clouds of dots (left and right) were presented on the screen for 200 ms. Participants were asked to indicate the side of the screen where they perceived more dots. Their response was kept on the screen for 1 s followed by feedback consisting of the symbolic number of dots in each cloud as well as the monetary gains and opportunity losses of the trial depending on the experimental condition. In the value condition, participants were explicitly informed that each dot in a cloud of dots corresponded to 1 Swiss Franc (CHF). Participants were informed that they would receive the amount in CHF corresponding to the total number of dots on the chosen side. At the end of the experiment a random trial was selected and they received the corresponding amount. In the accuracy condition, participants were explicitly informed that they could receive a fixed reward (15 Swiss Francs (CHF)) for each correct trial. This fixed amount was selected such that it approximately matched the expected reward received in the value condition (as tested in pilot experiments). At the end of the experiment, a random trial was selected and they would receive this fixed amount if they chose the cloud with more dots (i.e., the correct side). Each condition lasted for two consecutive days with the starting condition randomized across participants. Only after completing all four experiment days, participants were compensated for their time with 20 CHF per hour, in addition to the money obtained based on their decisions on each experimental day.

## Experiment 2

Participants (n = 6) carried out a numerosity discrimination task in which each of four daily sessions consisted of 16 runs of 40 trials each, thus each participant completed ~2560 trials. A key difference with respect to Experiment 1 is that participants had to accumulate points based on their decisions and had to reach a predetermined threshold on each run. The rules of point accumulation depended on the experimental condition. In the perceptual condition, a fixed amount of points was awarded if the participants chose the cloud with more dots. In this condition, participants were instructed to accumulate a number of points and reach a threshold given a limited number of trials. Based on the results obtained in Experiment 1, the threshold corresponded to 85% of correct trials in a given run, however the participants were unaware of this. If the participants reached this threshold, they were eligible for a fixed reward (20 CHF) as described in Experiment 1. In the value condition, the number of points received was equal to the number of dots in the cloud, however, contrary to Experiment 1, points were only awarded if the participant chose the cloud with the most dots. Participants had to reach a threshold that was matched in the expected collection of points of the perceptual condition. As in Experiment 1, each condition lasted for two consecutive days with the starting condition randomized across participants. Only after completing all the four days of the experiment, participants were compensated for their time with 20 CHF per hour, in addition to the money obtained based on their decisions on each experimental day.

## Experiment 3

The design of Experiment 3 was similar to the value condition of Experiment 2 (n = 7 participants) and was carried out over three consecutive days. The key difference between Experiment 3 and Experiments 1–2 was the shape of the prior distribution $f(v)$ that was used to draw the number of dots for each cloud in each trial (see below).

### Stimuli statistics and trial selection

For all experiments, we used the following parametric form of the prior distribution

$$f(v) = c(1-v)^{\alpha}, \tag{10}$$

initially defined in the interval [0,1] for mathematical tractability in the analytical solution of the encoding rules $\theta(v)$ (see below), with $\alpha > 0$ determining the shape of the distribution, and $c$ is a normalizing constant. For Experiments 1 and 2 the shape parameter was set to $\alpha = 2$, and for Experiment 3 was set to $\alpha = 1$. i.i.d. samples drawn from this distribution were then multiplied by 50,

added an offset of 5, and finally were rounded to the closest integer (i.e., the numerosity values in our experiment ranged from $v_{\min} = 5$ to $v_{\max} = 55$). The pairs of dots on each trial were determined by sampling from a uniform density window in the CDF space (*Equation 10* is its corresponding PDF). The pairs of dots in each trial were selected with the conditions that, first, their distance in the CDF space was less than a constant (0.25, 0.28 and 0.23 for Experiments 1, 2 and 3 respectively), and second, the number of dots in both clouds was different. *Figure 3c* illustrates the probability that a pair of choice alternatives was selected for a given trial in Experiments 1 and 2.

## Power analyses and model recovery

Given that adaptation dynamics in sensory systems often require long-term experience with novel prior distributions, we opted for maximizing the number of trials for a relatively small number of participants per experiment, as it is commonly done for this type of psychophysical experiments (*Brunton et al., 2013*; *Stocker and Simoncelli, 2006*; *Zylberberg et al., 2018*). Note that based on the power analyses described below, we collected in total ~45,000 trials across the three Experiments, which is above the average number of trials typically collected in human studies.

In order to maximize statistical power in the differentiation of the competing encoding rules, we generated 10,000 sets of experimental trials for each encoding rule and selected the sets of trials with the highest discrimination power (i.e., largest differences in Log-Likelihood) between the encoding models. In these power analyses, we also investigated what was the minimum number of trials that would allow accurate generative model selection at the individual level. We found that ~1000 trials per participant in each experimental condition would be sufficient to predict accurately (P>0.95) the true generative model. Based on these analyses, we decided to collect at least 1200 trials per participant and condition (perceptual and value) in each of the three experiments. Model recovery analyses presented in *Figure 3d* illustrate the result of our power analyses (see also *Figure 3—figure supplement 1*).

## Apparatus

Eyetracking (EyeLink 1000 Plus) was used to check the participants' fixation during stimulus presentation. When participants blinked or moved their gaze (more than 2° of visual angle) away from the fixation cross during the stimulus presentation, the trial was canceled (only 212 out of 45,600 trials were canceled, that is, <0.5% of the trials). Participants were informed when a trial was canceled and were encouraged not to do so as they would not receive any reward for this trial. A chinrest was used to keep the distance between the participants and the screen constant (55 cm). The task was run using Psychtoolbox Version 3.0.14 on Matlab 2018a. The diameter of the dots varied between 0.42° and 1.45° of visual angle. The center of each cloud was positioned 12.6° of visual angle horizontally from the fixation cross and had a maximum diameter of 19.6° of visual angle. Following previous numerosity experiments (*van den Berg et al., 2017*; *Izard and Dehaene, 2008*), either the average dot size or the total area covered by the dots was maintained constant in both clouds for each trial. The color of each dot (white or black) was randomly selected for each dot. Stimuli sets were different for each participant but identical between the two conditions.

## Encoding rules and model fits

The parametrization of the prior $f(v)$ (*Equation 10*) allows tractable analytical solutions of the encoding rules $\theta_A(v)$, $\theta_R(v)$ and $\theta_D(v)$, that correspond to Accuracy maximization, Reward maximization, and DbS, respectively:

$$\theta_{\mathrm{A}}(v) = \sin\left[\frac{\pi}{2}\left(1 - (1-v)^{\alpha+1}\right)\right]^2 \tag{11}$$

$$\theta_{\mathrm{R}}(v) = \sin\left[\frac{\pi}{2}\left(1 + (v-1)((1-v)^{\alpha})^{2/3}\right)\right]^2 \tag{12}$$

$$\theta_{\mathrm{D}}(v) = 1 - (1-v)^{\alpha+1} \tag{13}$$

Graphical representation of the respective encoding rules is shown in *Figure 3e* for Experiments 1 and 2. Given an encoding rule $\theta(v)$, we now define the decision rule. The goal of the decision

maker in our task is always to decide which of two input values $v_1$ and $v_2$ is larger. Therefore, the agent choses $v_1$ if and only if the internal readings $k_1 > k_2$. Following the definitions of expected value and variance of binomial variables, and approximating for large $n$ (see Appendix 2), the probability of choosing $v_1$ is given by

$$\text{P}_{\text{choose}v_1} \approx \Phi\left(\frac{\theta_1 - \theta_2}{\sqrt{\frac{\theta_1(1-\theta_1) + \theta_2(1-\theta_2)}{n}}}\right) \tag{14}$$

where $\Phi()$ is the standard CDF, and $\theta_1$ and $\theta_2$ are the encoding rules for the input values $v_1$ and $v_2$, respectively. Thus, the choice structure is the same for all models, only differing in their encoding rule. The three models generate different qualitative performance predictions for a given number of samples $n$ (*Figure 3f*).

Crucially, this probability decision rule (*Equation 14*) can be parsimoniously extended to include potential side biases independent of the encoding process as follows

$$\text{P}_{\text{choose}v_1} \approx \Phi\left(\frac{\theta_1 - \theta_2}{\sqrt{\frac{\theta_1(1-\theta_1) + \theta_2(1-\theta_2)}{n}}} + \beta_0\right) \tag{15}$$

where $\beta_0$ is the bias term. This is the base model used in our work. We were also interested in studying whether choice history effects (*Abrahamyan et al., 2016*; *Talluri et al., 2018*) may have influence in our task, thus possibly affecting the conclusions that can be drawn from the base model. Therefore, we extended this model to incorporate the effect of decision learning and choices from the previous trial

$$\text{P}_{\text{choose}v_1} \approx \Phi\left(\frac{\theta_1 - \theta_2}{\sqrt{\frac{\theta_1(1-\theta_1) + \theta_2(1-\theta_2)}{n}}} + \beta_0 + \beta^{\text{L}}a_{t-1}r_{t-1} + \beta^{\text{Ch}}a_{t-1}\right), \tag{16}$$

where $a_{t-1}$ is the choice made on the previous trial (+1 for left choice and $-1$ for right choice) and $r_{t-1}$ is the 'outcome learning' on the previous trial (+1 for correct choice and $-1$ for incorrect choice). $\beta^{\text{L}}$ and $\beta^{\text{Ch}}$ capture the effect of decision learning and choice in the previous trial, respectively.

Given that the choice structure is the same for all three sampling models considered here, we can naturally address the question of what decision rule the participants favor via a latent-mixture model. We implemented this model based on a hierarchical Bayesian modelling (HBM) approach. The base-rate probabilities for the three different encoding rules at the population level are represented by the vector $\pi$, so that $\pi_m$ is the probability of selecting encoding rule model $m$. We initialize the model with an uninformative prior given by

$$\pi \sim \text{Dirichlet}(1_{m=1}, 1_{m=2}, 1_{m=3}).$$

This base-rate is updated based on the empirical data, where we allow each participant $s$ to draw from each model categorically based on the updated base-rate

$$m_s \sim \text{Categorical}(\pi),$$

where the encoding rule $\theta$ for model $m$ is given by

$$\theta_{m,s} = \begin{cases} \theta_A, & m = 1 \\ \theta_R, & m = 2 \\ \theta_D, & m = 3 \end{cases}$$

The selected rule was then fed into *Equations 15 and 16* to determine the probability of selecting a cloud of dots. The number of samples $n$ was also estimated within the same HBM with population mean $\mu\mu$ and standard deviation $\sigma$ initialized based on uninformative priors with plausible ranges

$$\mu_n \sim \text{Uniform}(1, 1000)$$
$$\sigma_n \sim \text{Uniform}(0.01, 1000)$$

allowing each participant $s$ to draw from this population prior assuming that $n$ is normally distributed at the population level

$$n_s \sim \text{Normal}(\mu_n, \sigma_n)$$

Similarly, the latent variables $\beta$ in equations *Equations 15 and 16* were estimated by setting population mean $\mu_\beta$ and standard deviation $\sigma_\beta$ initialized based on uninformative priors

$$\mu_\beta \sim \text{Uniform}(-10, 10)$$
$$\sigma_\beta \sim \text{Uniform}(0.01, 100)$$

allowing each participant $s$ to draw from this population prior assuming that $\beta$ is normally distributed at the population level

$$\beta_s \sim \text{Normal}(\mu_\beta, \sigma_\beta)$$

In all the results reported in *Figure 3* and *Figure 4*, the value of the shape parameter of the prior was set to its true value $\alpha = 2$. The estimation of $\alpha$ in *Figure 5a* was investigated with a similar hierarchical approach, allowing each participant to sample from the normal population distribution with uninformative priors over the population mean and standard deviation

$$\mu_\alpha \sim \text{Uniform}(0.01, 20)$$
$$\sigma_\alpha \sim \text{Uniform}(0.0001, 100)$$

The choice rule of the standard logarithmic model of numerosity discrimination is given by

$$\text{P}_{\text{choose}v_1} = \Phi\left(\frac{\log(v_1) - \log(v_2)}{\sigma\sqrt{2}}\right), \tag{17}$$

where $\sigma$ is the internal noise in the logarithmic space. This model was extended to incorporate bias and choice history effects in the same way as implemented in the sampling models. Here, we emphasize that all sampling and log models have the same degrees of freedom, where performance is mainly determined by $n$ in the sampling models and Weber's fraction $\sigma$ in the log model, and biases are determined by parameters $\beta$. For all above-mentioned models, the trial-by-trial likelihood of the observed choice (i.e., the data) given probability of a decision was based on a Bernoulli process

$$y_{t,s} \sim \text{Bernoulli}(\text{P}_{\text{choose } v_1})$$

where $y_{t,s} \in \{0, 1\}$ is the decision of each participant $s$ in each trial $t$. In order to allow for prior adaptation, the model fits presented in *Figure 3* and *Figure 4* were fit starting after a fourth of the daily trials (corresponding to 150 trials for Experiment 1 and 160 trials for Experiment 2) to allow for prior adaptation and fixing the shape parameter to its true generative value $\alpha = 2$.

The dynamics of adaptation (*Figure 5*) were studied by allowing the shape parameter $\alpha$ to evolve through trial experience using all trials collected on each experiment day. This was studied using the following function

$$\alpha_t = \delta + \eta e^{-t/\tau}, \tag{18}$$

where $\delta$ represents a possible target adaptation value of $\alpha$, $t$ is the trial number, and $\eta$, $\tau$ determine the shape of the adaptation. Therefore, the encoding rule of the DbS model also changed trial-to-trial

$$\theta_{\text{D}}^t(v) = 1 - (1 - v)^{\alpha_t + 1}. \tag{19}$$

Adaptation was tested based on the hypothesis that participants initially use a logarithmic discrimination rule (*Equation 17*) (this strategy also allowed improving identification of the adaptation dynamics). Therefore, *Equation 18* was parametrized such that the initial value of the shape

parameter ($\alpha_{t=0}$) guaranteed that discriminability between the DbS and the logarithmic rule was as close as possible. This was achieved by finding the value of $\alpha$ in the DbS encoding rule ($\theta_D$) that minimizes the following expression

$$\sum_{t=1}^{T}\left[\left(\frac{\theta_D(v_{1,t})-\theta_D(v_{2,t})}{\sqrt{\theta_D(v_{1,t})(1-\theta_D(v_{1,t}))+\theta_D(v_{2,t})(1-\theta_D(v_{2,t}))}}\right)-\left(\log(v_{1,t})-\log(v_{2,t})\right)\right]^2, \quad (20)$$

where $v_{1,t}$ and $v_{2,t}$ are the numerosity inputs for each trial $t$. This expression was minimized based on all trials generated in Experiments 1–3 (note that minimizing this expression does not require knowledge of the sensitivity levels $\sigma$ and $n$ for the log and DbS models, respectively). We found that the shape parameter value that minimizes *Equation 20* is $\alpha=2.58$. Based on our prior $f(v)$ parametrization (*Equation 10*), this suggests that the initial prior is more skewed than the priors used in Experiments 1–3 (*Figure 5b*). This is an expected result given that log-normal priors, typically assumed in numerosity tasks, are also highly skewed. We fitted the $\delta$ parameter independently for Experiments 1–2 and Experiment 3 but kept the $\tau$ parameter shared across all experiments. If adaptation is taking place, we hypothesized that the asymptotic value $\delta$ of the shape parameter $\alpha$ should be larger for Experiments 1–2 compared to Experiment 3.

Posterior inference of the parameters in all the hierarchical models described above was performed via the Gibbs sampler using the Markov Chain Monte Carlo (MCMC) technique implemented in JAGS. For each model, a total of 50,000 samples were drawn from an initial burn-in step and subsequently a total of new 50,000 samples were drawn for each of three chains (samples for each chain were generated based on a different random number generator engine, and each with a different seed). We applied a thinning of 50 to this final sample, thus resulting in a final set of 1000 samples for each chain (for a total of 3000 pooling all three chains). We conducted Gelman–Rubin tests for each parameter to confirm convergence of the chains. All latent variables in our Bayesian models had $\hat{R}<1.05$, which suggests that all three chains converged to a target posterior distribution. We checked via visual inspection that the posterior population level distributions of the final MCMC chains converged to our assumed parametrizations. When evaluating different models, we are interested in the model's predictive accuracy for unobserved data, thus it is important to choose a metric for model comparison that considers this predictive aspect. Therefore, in order to perform model comparison, we used a method for approximating leave-one-out cross-validation (LOO) that uses samples from the full posterior (*Vehtari et al., 2017*). These analyses were repeated using an alternative Bayesian metric: the WAIC (*Vehtari et al., 2017*).

## Acknowledgements

This work was supported by an ERC starting grant (ENTRAINER) to RP and by a grant of the U.S. National Science Foundation to M.W. This project has received funding from the European Research Council (ERC) under the European Union's Horizon 2020 research and innovation programme (grant agreement No. 758604).

## Additional information

### Funding

| Funder | Grant reference number | Author |
|---|---|---|
| European Commission | grant agreement No. 758604 | Rafael Polania |
| ERC | ENTRAINER | Rafael Polania |
| National Science Foundation | | Michael Woddford |

The funders had no role in study design, data collection and interpretation, or the decision to submit the work for publication.

## Author contributions
Joseph A Heng, Conceptualization, Software, Formal analysis, Investigation, Visualization, Methodology, Writing - original draft, Writing - review and editing; Michael Woodford, Conceptualization, Formal analysis, Writing - original draft, Writing - review and editing; Rafael Polania, Conceptualization, Resources, Formal analysis, Supervision, Funding acquisition, Investigation, Writing - original draft, Project administration, Writing - review and editing

## Author ORCIDs
Joseph A Heng (ID) https://orcid.org/0000-0002-3643-4623
Rafael Polania (ID) https://orcid.org/0000-0002-6176-6806

## Ethics
Human subjects: Informed consent, and consent to publish, was obtained. The experiments conformed to the Declaration of Helsinki and the experimental protocol was approved by the Ethics Committee of the Canton of Zurich (BASEC: 2018-00659).

## Decision letter and Author response
Decision letter https://doi.org/10.7554/eLife.54962.sa1
Author response https://doi.org/10.7554/eLife.54962.sa2

# Additional files
## Supplementary files
• Transparent reporting form

## Data availability
Data and essential code that support the findings of this study have been made available at the Open Science Framework (https://osf.io/xgfu9/).

The following dataset was generated:

| Author(s) | Year | Dataset title | Dataset URL | Database and Identifier |
|---|---|---|---|---|
| Heng JA | 2020 | Efficient sampling and noisy decisions | https://osf.io/xgfu9/ | Open Science Framework, xgfu9 |

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

## Appendix 1

### Infomax coding rule

We assume that the subjective perception of an environmental variable with value $v$ is determined by $n$ independent *samples* of a binary random variable, that is, outcomes are either 'high' (ones) or 'low' (zeros) readings. Here, the probability $\theta$ of a 'high' reading is the same on each draw, but can depend on the input stimulus value, via the function $\theta(v)$. Additionally, we assume that the input value $v$ on a given trial is an independent draw from some prior distribution $f(v)$ in a given environment or context (with $F(v)$ being the corresponding cumulative distribution function). As we mentioned before, the choice of $\theta$ (i.e., encoding of the input value) depends on $v$. Now suppose that the mapping $\theta(v)$ (the encoding rule) is chosen so as to maximize the mutual information between the random variable $v$ and the subjective value representation $k$. The mutual information is computed under the assumption that $v$ is drawn from a particular prior distribution $f(v)$, and $\theta(v)$ is assumed to be optimized for this prior. The mutual information between $v$ and $k$ is defined as

$$I(v,k) = H(k) - H(k|v), \tag{21}$$

where the marginal entropy $H(k)$ quantifies the uncertainty of the marginal response distribution $P(k)$, and $H(k|v)$ is the average conditional entropy of $k$ given $v$. The output distribution is given by

$$P(k) = \int_{v \in V} P(k|v) f(v) dv, \tag{22}$$

where $f(v)$ is defined as the input density function. For the encoding framework that we consider here, which is given by the binomial channel, the conditional probability mass function of the output given the input is

$$P(k|v) = nk\theta(v)^k (1 - \theta(v))^{n-k}, \quad k \in [0, 1, \ldots, n]. \tag{23}$$

Thus, we have all the ingredients to write the expression of the mutual information

$$\begin{aligned} I(v,k) \ &= H(k) - H(k|v) \\ &= -\sum_{k=0}^{n} P(k) \log P(k) \\ &\quad - \left( -\int_{v \in V} f(v) \sum_{k=0}^{n} P(k|v) \log P(k|v) \ dv \right) \end{aligned} \tag{24}$$

We then seek to determine the encoding rule $\theta(v)$ that solves the optimization problem

$$\text{find} \quad C = \max_{\{\theta(v)\}} I(v,k). \tag{25}$$

It can be shown that for large $n$, the mutual information between $\theta$ and $k$ (hence the mutual information between $v$ and $k$) is maximized if the prior distribution over $\theta$ is the Jeffreys prior (***Clarke and Barron, 1994***)

$$\text{Beta}(\theta; 0.5, 0.5) = \frac{1}{\pi \sqrt{\theta(1-\theta)}}, \tag{26}$$

also known as the arcsine distribution. Hence, the mapping $\theta(v)$ induces a prior distribution over $\theta$ given by the arcsine distribution. This means that for each $v$, the encoding function $\theta(v)$ must be such that

$$\begin{aligned} F(v) \ &= \int_{0}^{\theta(v)} \frac{1}{\pi \sqrt{\tilde{\theta}(1-\tilde{\theta})}} d\tilde{\theta} \\ &= \frac{2}{\pi} \arcsin(\sqrt{\theta(v)}). \end{aligned} \tag{27}$$

Solving for $\theta$ we finally obtain the optimal encoding rule

$$\theta(v) = \left[\sin\left(\frac{\pi}{2}F(v)\right)\right]^2.$$

## Appendix 2

### Accuracy maximization for a known prior distribution

Here we derive the optimal encoding rule when the criterion to be maximized is the probability of a correct response in a binary comparison task, rather than mutual information as in Appendix 1. As in Appendix 1, we assume that the prior distribution $f$ from which stimuli are drawn is known, and that the encoding rule is optimized for this particular distribution. (The case in which we wish the encoding rule to be robust to variations in the distribution from which stimuli are drawn is instead considered in Appendix 6.) Note that the objective assumed here corresponds to maximization of expected reward in the case of a perceptual experiment in which a subject must indicate which of two presented magnitudes is greater, and is rewarded for the number of correct responses. (In Appendix 5, we instead consider the encoding rule that would maximize expected reward if the subject's reward is proportional to the magnitude selected by their response).

As above, we assume encoding by a binomial channel. The encoded value (number of 'high' readings) is given by $k$, which is consequently an integer between 0 and $n$. This is a random variable with a binomial distribution with expected value and variance given by

$$\mathrm{E}\left[\frac{k}{n}|\theta\right] = \theta \qquad \mathrm{Var}\left[\frac{k}{n}|\theta\right] = \frac{\theta(1-\theta)}{n} \tag{29}$$

Suppose that the task of the decision maker is to decide which of two input values $v_1$ and $v_2$ is larger. Assuming that $v_1$ and $v_2$ are encoded independently, then the decision maker choses $v_1$ if and only if the internal readings $k_1 > k_2$ (here we may suppose that the probability of choosing stimulus 1 is 0.5 in the event that $k_1 = k_2$). Thus, the probability of choosing stimulus 1 is:

$$\mathrm{P}\left(\frac{k_1}{n} > \frac{k_2}{n}|v_1, v_2\right) + \frac{1}{2}\mathrm{P}\left(\frac{k_1}{n} = \frac{k_2}{n}|v_1, v_2\right). \tag{30}$$

In the case of large $n$, we can use a normal approximation to the binomial distribution to obtain

$$\left(\frac{k_1}{n} - \frac{k_2}{n}\right) \sim \mathrm{N}\left(\theta_1 - \theta_2, \frac{\theta_1(1-\theta_1) + \theta_2(1-\theta_2)}{n}\right) \tag{31}$$

and hence the probability of choosing $v_1$ is given by

$$\mathrm{P}_{\mathrm{choose}\,v_1} \approx \Phi\left(\frac{\theta_1 - \theta_2}{\sqrt{\frac{\theta_1(1-\theta_1) + \theta_2(1-\theta_2)}{n}}}\right), \tag{32}$$

where $\Phi(\cdot)$ is the standard CDF. Thus the probability of an incorrect choice (i.e., choosing the item with the lower value) is approximately

$$\mathrm{P}_{\mathrm{error}} \approx \Phi\left(-\frac{|\theta_1 - \theta_2|}{\sqrt{\frac{\theta_1(1-\theta_1) + \theta_2(1-\theta_2)}{n}}}\right) \tag{33}$$

Now, suppose that the encoding rule, together with the prior distribution for $v$ (the same for both inputs that are independent draws from the prior distribution) results in an ex-ante distribution for $\theta$ (same for both goods) with density function $\hat{f}(\theta)$. Then the probability of error is given by

$$\mathrm{P}_{\mathrm{error}} \approx \int \int \Phi\left(-\frac{|\theta_1 - \theta_2|}{\sqrt{\frac{\theta_1(1-\theta_1) + \theta_2(1-\theta_2)}{n}}}\right) \hat{f}(\theta_1)\hat{f}(\theta_2)\, d\theta_1 d\theta_2 \tag{34}$$

Our goal is to evaluate *Equation 34* for any choice of the density $\hat{f}(\theta)$. First, we fix the value of $\theta_1$ and integrate over $\theta_2$:

$$\int_0^1 \Phi\left(-\frac{|\theta_1-\theta_2|}{\sqrt{\theta_1(1-\theta_1)+\theta_2(1-\theta_2)}}\sqrt{n}\right)\hat{f}(\theta_2)d\theta_2$$

$$= \int_0^{\theta_1} \Phi\left(-\frac{\theta_2-\theta_1}{\sqrt{\theta_1(1-\theta_1)+\theta_2(1-\theta_2)}}\sqrt{n}\right)\hat{f}(\theta_2)d\theta_2 \qquad (35)$$

$$+ \int_{\theta_1}^1 \Phi\left(-\frac{\theta_1-\theta_2}{\sqrt{\theta_1(1-\theta_1)+\theta_2(1-\theta_2)}}\sqrt{n}\right)\hat{f}(\theta_2)d\theta_2$$

with $\theta_2 = \theta_1 + \sqrt{2n\theta_1(1-\theta_1)}z$, the expression above then becomes

$$\approx \int_{\frac{-\theta_1\sqrt{n}}{\sqrt{2\theta_1(1-\theta_1)}}}^0 \Phi(z)\hat{f}(\theta_1)\left[\frac{\sqrt{2\theta_1(1-\theta_1)}}{\sqrt{n}}\right]dz$$

$$+ \int_0^{\frac{(1-\theta_1)\sqrt{n}}{\sqrt{2\theta_1(1-\theta_1)}}} \Phi(-z)\hat{f}(\theta_1)\left[\frac{\sqrt{2\theta_1(1-\theta_1)}}{\sqrt{n}}\right]dz \qquad (36)$$

$$\approx \underbrace{\left[2\int_{-\infty}^0 \Phi(z)dz\right]}_{>0}\hat{f}(\theta_1)\frac{\sqrt{2\theta_1(1-\theta_1)}}{\sqrt{n}}$$

Then we can integrate over $\theta_1$ to obtain:

$$\mathrm{P_{error}} \approx \frac{2}{\sqrt{n\pi}}\int \hat{f}(\theta_1)^2\sqrt{(\theta_1(1-\theta_1))}\,d\theta_1. \qquad (37)$$

This problem can be solved using the method of Lagrange multipliers:

$$\int \sqrt{\theta(1-\theta)}\hat{f}(\theta)^2 d\theta + \lambda\left(\int \hat{f}(\theta)-1\right)$$

$$= \int (\sqrt{\theta(1-\theta)}\hat{f}(\theta)^2 + \lambda\hat{f}(\theta))d\theta - \lambda$$

$$= \int \mathcal{L}(\theta,\hat{f},\lambda)d\theta - \lambda$$

We now calculate the gradient

$$\frac{\partial\mathcal{L}}{\partial\hat{f}} = 2\hat{f}\sqrt{(\theta(1-\theta))} + \lambda \qquad (39)$$

and then find the optimum for $\hat{f}$ by setting

$$2\hat{f}\sqrt{(\theta(1-\theta))} + \lambda = 0 \qquad (40)$$

then solving for $\hat{f}$ to obtain

$$\hat{f} = \frac{-\lambda}{2\sqrt{\theta(1-\theta)}}. \qquad (41)$$

Taken into consideration our optimization constraint, it can be shown that

$$\int_0^1 \frac{1}{\sqrt{\theta(1-\theta)}} = \frac{1}{\pi}$$

and therefore this implies:

$$\frac{1}{\pi} = \frac{-\lambda}{2}$$

thus requiring:

$$-\lambda = \frac{2}{\pi}.$$

Replacing $\lambda$ in *Equation 41* we finally obtain

$$\hat{f}(\theta) = \frac{1}{\pi\sqrt{\theta(1-\theta)}} \qquad \text{(26 revisited)}$$

Thus the optimal encoding rule is the same (at least in the large-$n$ limit) in this case as when we assume an objective of maximum mutual information (the case considered in Appendix 1), though here we assume that the objective is accurate performance of a specific discrimination task.

# Appendix 3

## Optimal noise for a known prior distribution

Interestingly, we found that the fundamental principles of the theory independently developed in our work are directly linked to the concept of suprathreshold stochastic resonance (SSR) discovered about two decades ago. Briefly, SSR occurs in an array of $n$ identical threshold non-linearities, each of which is subject to independently sampled random additive noise (**Figure 1** in main text). SSR should not be confused with the standard stochastic resonance (SR) phenomenon. In SR, the amplitude of the input signal is restricted to values smaller than the threshold for SR to occur. On the other hand, in SSR, random draws from the distribution of input values can exist above threshold levels. Using the simplified implementational scheme proposed in our work, it can be shown that mutual information $I(v, k)$ can be also optimized by finding the optimal noise distribution. This is important as it provides a normative justification as for why sampling must be noisy in capacity-limited systems. Actually, SSR was initially motivated as a model of neural arrays such as those synapsing with hair cells in the inner ear, with the direct application of establishing the mechanisms by which information transmission can be optimized in the design of cochlear implants (**Stocks et al., 2002**). Our goal in this subsection is to make evident the link between the novel theoretical implications of our work and the SSR phenomenon developed in previous work (**Stocks et al., 2002**; **McDonnell et al., 2007**), which should further justify our argument of efficient noisy sampling as a general framework for decision behavior, crucially, with a parsimonious implementational nature.

Following our notation, each threshold device (we will call it from now on a *neuron*) can be seen as the number of $n$ resources available to encode an input stimulus $v$. Here, we assume that each neuron produces a 'high' reading if and only if $v + \eta > \tau$, where $\eta$ is i.i.d. random additive noise (independent of $v$) following a distribution function $f_\eta$, and $\tau$ is the minimum threshold required to produce a 'high' reading. If we define the noise CDF as $F_\eta$, then the probability $\theta$ of the neuron giving a 'high' reading in response to the input signal $v$ is given by

$$\theta(v) = 1 - F_\eta(\tau - v). \tag{42}$$

It can be shown that the mutual information between the input $v$ and the number of 'high' readings $k$ for large $n$ is given by **McDonnell et al., 2007**,

$$I(v, k) \approx \frac{1}{2} \log_2 \left( \frac{n\pi}{2e} \right) - D_{\mathrm{KL}}[f(v) \| f_J(v)], \tag{43}$$

where $f_J$ is the Jeffreys prior (**Equation 26**). Therefore, Jeffreys' prior can also be derived making it a function of the noise distribution $f_\eta$

$$f_J(v) = \frac{f_\eta(\tau - v)}{\pi \sqrt{F_\eta(\tau - v)[1 - F_\eta(\tau - v)]}}. \tag{44}$$

Given that the first term in **Equation 43** is always non-negative, a sufficient condition for achieving channel capacity is given by

$$f(v) = f_J(v) \quad \forall v. \tag{45}$$

Typically, the nervous system of any organism has little influence on the distribution of physical signals in the environment. However, it has the ability to shape its internal signals to optimize information transfer. Therefore, a parsimonious solution that the nervous system may adopt to adapt to statistical regularities of environmental signals in a given context is to find the optimal noise distribution $f_\eta^*$ to achieve channel capacity. Note that this is different from classical problems in communication theory where the goal is usually to find the signal distribution that maximizes mutual information for a channel. Solving **Equation 44** to find $f_\eta(v)$ one can find such optimal noise distribution

$$f_\eta^*(v) = \frac{\pi}{2} \sin[\pi(1 - F(\tau - v))] f(\tau - v). \tag{46}$$

A further interesting consequence of this set of results is that the ratio between the signal PDF $f(v)$ and the noise PDF $f_\eta$ is

$$\frac{f(v)}{f_\eta(\tau - v)} = \frac{2}{\pi \sin[\pi(1 - F(v))]}. \tag{47}$$

Using the definition given in *Equation 42* to make this expression a function of $\theta$, one finds the optimal PDF of the encoder

$$f^*(\theta) = \frac{1}{\pi \sqrt{\theta(1 - \theta)}}, \tag{48}$$

which is once again the arcsine distribution (See *Equations 2 and 5* in main text).

## Appendix 4

### Efficient coding and the relation between environmental priors and discrimination

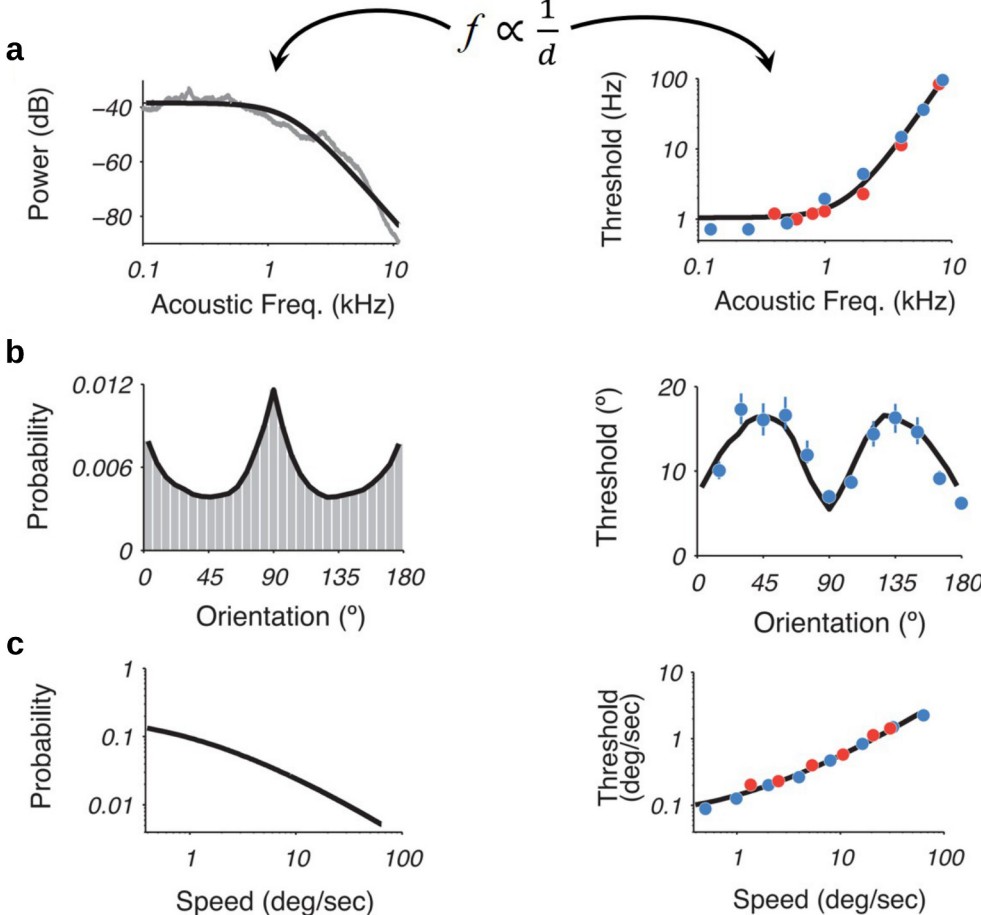

**Appendix 4—figure 1.** Recently, it was shown that using an efficiency principle for encoding sensory variables, based on population of noisy neurons, it was possible to obtain an explicit relationship between the statistical properties of the environment (the prior) and perceptual discriminability (*Ganguli and Simoncelli, 2016*). The theoretical relation states that discriminability should be inversely proportional to the density of the prior distribution. Interestingly, this relationship holds across several sensory modalities such as (**a**) acoustic frequency, (**b**) local orientation, (**c**) speed (figure adapted with permission from the authors *Ganguli and Simoncelli, 2016*). Here, we investigate whether this particular relation also emerges in our efficient sampling framework.

We first show that we obtain a prediction of exactly the same kind from our model of encoding using a binary channel, in the case that (i) we assume that the encoding rule is optimized for a single environmental distribution, as in the theory of *Ganguli and Simoncelli, 2014*; *Ganguli and Simoncelli, 2016*, and (ii) the objective that is maximized is either mutual information (as in the theory of Ganguli and Simoncelli) or the probability of an accurate binary comparison (as considered in Appendix 2).

Note that the expected value and variance of a binomial random variable are given by

$$\mathrm{E}[r|\theta] = \theta \quad \mathrm{Var}[r|\theta] = \frac{\theta(1-\theta)}{n},\tag{49}$$

where we let here $r \equiv k/n$. In Appendix 2, we show that if the objective is accuracy maximization, an efficient binomial channel requires that

$$\theta(v) = \left[\sin\left(\frac{\pi}{2}F(v)\right)\right]^2.$$

Thus, replacing $\theta(v)$ in *Equation 49* implies the following relations

$$\mathrm{E}[r|\theta] = \sin^2(\omega), \qquad \mathrm{Var}[r|\theta] = \frac{\sin^2(\omega)\cos^2(\omega)}{n}, \tag{50}$$

where we let here $\omega \equiv \frac{\pi}{2}F(v)$. Discrimination thresholds $d$ in sensory perception are defined as the ratio between the precision of the representation and the rate of change in the perceived stimulus

$$d \equiv \frac{\sqrt{\mathrm{Var}[r|\theta]}}{\mathrm{E}[r|\theta]'}. \tag{51}$$

Substituting the expressions for expected value and variance in *Equation 50* results in

$$\begin{aligned} d \ &= \frac{1}{2\sqrt{n}\omega'} \\ &= \frac{1}{\sqrt{n}\pi f(v)}. \end{aligned} \tag{52}$$

Thus under our theory, this implies

$$d \propto \frac{1}{f(v)}. \tag{53}$$

This is exactly the relationship derived and tested by *Ganguli and Simoncelli, 2016*.

Our model instead predicts a somewhat different relationship if the encoding rule is required to be robust to alternative possible environmental frequency distributions (the case further discussed in Appendix 6). In this case, the robustly optimal encoding rule is DbS, which corresponds to $\theta(v) = F(v)$, rather than the relation *(53)*. Substituting this into *Equations 49 and 51* yields the prediction

$$d = \frac{\sqrt{F(v)(1 - F(v))}}{\sqrt{n}} \cdot \frac{1}{f(v)}. \tag{54}$$

instead of *Equation 52*.

One interpretation of the experimental support for relation *(53)* reviewed by *Ganguli and Simoncelli, 2016* could be that in the case of early sensory processing of the kind with which they are concerned, perceptual processing is optimized for a particular environmental frequency distribution (representing the long-run experience of an organism or even of the species), so that the assumptions used in Appendix 2 are the empirically relevant ones. Even so, it is arguable that robustness to changing contextual frequency distributions should be important in the case of higher forms of cognition, so that one might expect prediction of *Equation 54* to be more relevant for these cases; and indeed, our experimental results for the case of numerosity discrimination are more consistent with *Equation 54* than with *Equation 52*.

One should also note that even in a case where *Equation 54* holds, if one measures discrimination thresholds over a subset of the stimulus space, over which there is non-trivial variation in $f(v)$, but $F(v)$ does not change very much (because the prior distribution for which the encoding rule is optimized assigns a great deal of probability to magnitudes both higher and lower than those in the experimental data set), then relation *(54)* restricted to this subset of the possible values for $v$ will imply that relation *(53)* should approximately hold. This provides another possible interpretation of the fact that the relation *(53)* holds fairly well in the data considered by *Ganguli and Simoncelli, 2016*.

# Appendix 5

## Maximizing expected size of the selected item (fitness maximization)

We now consider the optimal encoding rule under a different assumed objective, namely, maximizing the expected magnitude of the item selected by the subject's response (that is, the stimulus judged to be larger by the subject), rather than maximizing the probability of a correct response as in Appendix 2. While in many perceptual experiments, maximizing the probability of a correct response would correspond to maximization of the subject's expected reward (or at least maximization of a psychological reward to the subject, who is given feedback about the correctness of responses but not about true magnitudes), in many of the ecologically relevant cases in which accurate discrimination of numerosity is useful to an organism (*Butterworth et al., 2018*; *Nieder, 2020*), the decision maker's reward depends on how much larger one number is than another, and not simply their ordinal ranking. This would also be true of typical cases in which internal representations of numerical magnitudes must be used in economic decision making: the reward from choosing an investment with a larger monetary payoff is proportional to the size of the payoff afforded by the option that is chosen. Hence it is of interest to consider the optimal encoding rule if we suppose that encoding is optimized to maximize performance in a decision task with this kind of reward structure.

As in Appendix 1 and Appendix 2, we again consider the problem of optimizing the encoding rule for a specific prior distribution $f(v)$ for the magnitudes that may be encountered, and we assume that it is only possible to encode information via 'high' or 'low' readings. The optimization problem that we need to solve is to find the optimal encoding function $\theta(v)$ that guarantees a maximal expected value of the chosen outcome, for any given prior distribution $f(v)$. Thus the quantity that we seek to maximize is given by

$$\mathrm{E}[v(\mathrm{chosen})] = \int\int f(v_1, v_2)\left[P_1(\theta(v_1), \theta(v_2))v_1 + P_2(\theta(v_1), \theta(v_2))v_2\right] dv_1 dv_2 \tag{55}$$

where $P_i(\theta_1, \theta_2)$ is the probability of choosing option $i$ when the encoded values of the two options are $\theta_1$ and $\theta_2$ respectively.

We begin by noting that for any pair of input values $v_1, v_2$, the integrand in *Equation 55* can be written as

$$\begin{aligned}
P_1 &(\theta(v_1), \theta(v_2))v_1 + P_2(\theta(v_1), \theta(v_2))v_2 \\
&= \max(v_1, v_2) - P_1(\theta(v_1), \theta(v_2))\max(v_2 - v_1, 0) - P_2(\theta(v_1), \theta(v_2))\max(v_1 - v_2, 0) \\
&= \max(v_1, v_2) - [P_1(\theta(v_1), \theta(v_2))I(v_2 > v_1) + P_2(\theta(v_1), \theta(v_2))I(v_1 > v_2)]|v_1 - v_2| \\
&= \max(v_1, v_2) - [P(\mathrm{error}|\theta(v_1), \theta(v_2))I(v_2 > v_1) + P(\mathrm{error}|\theta(v_1), \theta(v_2))I(v_1 > v_2)]|v_1 - v_2| \\
&= \max(v_1, v_2) - P(\mathrm{error}|\theta(v_1), \theta(v_2))|v_1 - v_2|,
\end{aligned} \tag{56}$$

where $I(A)$ is the indicator function (taking the value 1 if statement $A$ is true, and the value 0 otherwise), and $P(error|\theta_1, \theta_2)$ is the probability of choosing the lower-valued of the two options.

Substituting this last expression for the integrand in *Equation 55*, we see that we can equivalently write

$$\mathrm{E}[v(\mathrm{chosen})] = \mathrm{E}[\max(v_1, v_2)] - \int\int f(v_1, v_2)P(\mathrm{error}|\theta(v_1), \theta(v_2))|v_1 - v_2| dv_1 dv_2, \tag{57}$$

where

$$\mathrm{E}[\max(v_1, v_2)] \equiv \int\int f(v_1, v_2)\max(v_1, v_2) dv_1 dv_2 \tag{58}$$

is a quantity which is independent of the encoding function $\theta(v)$. Hence choosing $\theta(v)$ to maximize *Equation 55* is equivalent to choosing it to minimize

$$\mathrm{E}[\mathrm{loss}] = \int\int f(v_1, v_2)P(\mathrm{error}|\theta(v_1), \theta(v_2))|v_1 - v_2| dv_1 dv_2. \tag{59}$$

As previously specified, the probability of error given two internal noisy readings $k_1$ and $k_2$ is given by

$$P(\text{error}) = \left(\frac{k_1}{n} - \frac{k_2}{n} > 0 | v_1, v_2\right) \tag{60}$$

$$\approx \Phi\left(\frac{\theta_1 - \theta_2}{\sqrt{\frac{\theta_1(1-\theta_1) + \theta_2(1-\theta_2)}{n}}}\right), \tag{61}$$

where in this case we assume that $v_1$ is the lower-valued option and $v_2$ is the higher-valued option on any given trial. This implies that $P(\text{error})$ is very close to zero, except when $|\theta_1 - \theta_2| = \mathcal{O}(1/\sqrt{n})$. In this case we have

$$P(\text{error}) \approx \Phi\left(\sqrt{\frac{n}{2}} \frac{\theta_1 - \theta_2}{\sqrt{\theta(1-\theta)}}\right) \quad \text{where} \quad \theta \equiv \frac{\theta_1 + \theta_2}{2}. \tag{62}$$

As in the case of accuracy maximization, here we assume that $(v_1, v_2)$ are independent draws from the same distribution of possible values $f(v)$. Thus $f(v_1, v_2) = f(v_1)f(v_2)$. Then fixing $v_1$ and integrating over all possible values of $v_2$ in *Equation 59*, the expected loss is approximately

$$\mathrm{E}[\text{loss}|v_1] = \int f(v_2)P(\text{error}|v_2, v_1)|v_2 - v_1| dv_2 \tag{63}$$

$$\approx \int f(v_2)\Phi\left(-\sqrt{\frac{n}{2}} \frac{|\theta_1 - \theta_2|}{\sqrt{\theta_1(1-\theta_1)}}\right)|v_2 - v_1| dv_2 \tag{64}$$

$$\approx f(v_1) \int \Phi\left(-\sqrt{\frac{n}{2}} \frac{\theta'(v_1)|v_2 - v_1|}{\sqrt{\theta_1(1-\theta_1)}}\right)|v_2 - v_1| dv_2 \tag{65}$$

$$\approx f(v_1) \int_{-\infty}^{\infty} \Phi(-|z|) \left[\sqrt{\frac{2\theta_1(1-\theta_1)}{n}} \frac{|z|}{\theta'(v_1)}\right]\left[\sqrt{\frac{2\theta_1(1-\theta_1)}{n}} \frac{1}{\theta'(v_1)}\right] dz \tag{66}$$

$$\approx \frac{4}{n} \frac{f(v_1)}{\theta'(v_1)^2}[\theta_1(1-\theta_1)] \underbrace{\int_0^{\infty} \Phi(-z)z dz}_{1/4} \tag{67}$$

$$\approx \frac{1}{n} \frac{f(v_1)}{\theta'(v_1)^2}[\theta_1(1-\theta_1)] \tag{68}$$

where in *Equation 66* we have applied the change of variable

$$z \equiv \frac{n}{2} \frac{\theta'(v_1)}{\theta_1(1-\theta_1)}(v_2 - v_1) \tag{69}$$

and in the integral of *Equation 67* we have used

$$\int_0^{\infty} \Phi(-z)z dz = \frac{1}{2}\left[(z^2 - 1)\Phi(-z) - z\phi(-z)\right]_0^{\infty} \tag{70}$$

$$= \frac{1}{2}\left[0 - (-\frac{1}{2})\right] \tag{71}$$

$$= \frac{1}{4} \tag{72}$$

where $\phi()$ is the standard normal PDF. Then integrating over $v_1$, we have:

$$\mathrm{E}[\text{loss}] = \frac{1}{n} \int \frac{f(v_1)^2}{\theta'(v_1)^2}[\theta_1(1-\theta_1)] \, dv_1. \tag{73}$$

Thus we want to find the encoding rule $\theta(v)$ to minimize this integral given the prior $f(v)$. We now apply the change of variable $\theta(v) \equiv \sin^2(\gamma(v))$, where $\gamma(v)$ is an increasing function with a range $0 \leqslant \gamma(v) \leqslant \frac{\pi}{2}$ for all $v$. Then we have

$$\theta'(v) = 2\sin(\gamma(v))\cos(\gamma(v))\gamma'(v) \tag{74}$$

$$= 2\sqrt{\theta(v)(1-\theta(v))}\gamma'(v) \tag{75}$$

and therefore we have

$$\frac{\theta(v)(1-\theta(v))}{\theta'(v)} = \frac{1}{4}\frac{1}{\gamma'(v)}. \tag{76}$$

This allows us to rewrite *Equation 73* as follows

$$\mathrm{E}[\text{loss}] = \frac{1}{n}\int \frac{f(v)^2}{\gamma'(v)^2}. \tag{77}$$

Now the problem is to choose the function $\gamma(v)$ to minimize $\mathrm{E}[\text{loss}]$ subject to $0 \leqslant \gamma(v) \leqslant \frac{\pi}{2}$. Equivalently, we can choose the function $\gamma'(v) > 0$ to minimize $\mathrm{E}[\text{loss}]$ subject to $\int \gamma'(v)dv \leqslant \frac{\pi}{2}$. Defining $\varphi(v) \equiv \gamma'(v)$, the optimization problem to solve is to choose the function $\varphi(v)$ to

$$\min \quad \int \frac{f(v)^2}{\varphi(v)^2}dv \quad \text{s.t.} \quad \int \varphi(v)\,dv \leqslant \frac{\pi}{2} \tag{78}$$

Due to FOC, it can be shown that

$$\frac{f(v)^2}{\varphi(v)^3} = \text{same for all } v \quad \Rightarrow \quad \varphi(v) \sim f(v)^{2/3}. \tag{79}$$

Note also that the constraint $\int \varphi(v) \leqslant \frac{\pi}{2}$ must hold with equality, thus arriving at

$$\gamma(v) = \frac{\pi}{2}\frac{\displaystyle\int_{-\infty}^{v} f(\tilde{v})^{2/3}\,d\tilde{v}}{\displaystyle\int_{-\infty}^{\infty} f(\tilde{v})^{2/3}\,d\tilde{v}}. \tag{80}$$

Therefore, we finally obtain the efficient encoding rule that maximizes the expected magnitude of the selected item

$$\theta(v) = \sin\left[\frac{\pi}{2}\frac{\displaystyle\int_{-\infty}^{v} f(\tilde{v})^{2/3}\,d\tilde{v}}{\displaystyle\int_{-\infty}^{\infty} f(\tilde{v})^{2/3}\,d\tilde{v}}\right]^2 \tag{81}$$

# Appendix 6

## Robust optimality of DbS among encoding rules with $m = 1$

Here we consider the nature of the optimal encoding function when the cost of increasing the size of the sample of values from prior experience that are used to adjust the encoding rule to the contextual distribution of stimulus values is great enough to make it optimal to base the encoding of a new stimulus magnitude $v$ on a single sampled value $\tilde{v}$ from the contextual distribution. (The conditions required for this to be the case are discussed further in Appendix 7).

We assume that for each of the $n$ independent processing units, the probability of a 'high' reading is given by $\theta(v, \tilde{v}_j)$, where $\tilde{v}_j$ is the draw from the contextual distribution by processor $j$, and $\theta(v, \tilde{v})$ is the same function for each of the processing units. The $\{\tilde{v}_j\}$ for $j = 1, 2, \ldots, n$ are independent draws from the contextual distribution $f(v)$. We further assume that the function $\theta(v, \tilde{v})$ satisfies certain regularity conditions. First, we assume that $\theta$ is a piecewise continuous function. That is, we assume that the $v - \tilde{v}$ plane can be divided into a countable number of connected regions, with the boundaries between regions defined by continuous curves; and that the function $\theta(v, \tilde{v})$ is continuous in the interior of any of these regions, though it may be discontinuous at the boundaries between regions. And second, we assume that $\theta(v, \tilde{v})$ is necessarily weakly increasing in $v$ and weakly decreasing in $\tilde{v}$. The function is otherwise unrestricted.

For any prior distribution $f(v)$ and any encoding function $\theta(v, \tilde{v})$, we can compute the probability of an erroneous comparison when two stimulus magnitudes $v_1, v_2$ are independently drawn from the distribution $f(v)$, and each of these stimuli is encoded using $n$ additional independent draws $\{\tilde{v}_j\}$ from the same distribution. Let this error probability be denoted $P_n(\theta; f)$. We wish to find an encoding rule (for given $n$) that will make this error probability as small as possible; however, the answer to this question will depend on the prior distribution $f(v)$. Hence we wish to find an encoding rule that is robustly optimal, in the sense that it achieves the minimum possible value for the upper bound

$$\bar{P}_{error}(\theta) \equiv \sup_{f \in \mathcal{F}} P_n(\theta; f)$$

for the probability of an erroneous comparison. Here, the class of possible priors $\mathcal{F}$ to consider is the set of all possible probability distributions (over values of $v$) that can be characterized by an integrable probability density function $f(v)$. (We exclude from consideration priors in which there is an atom of probability mass at some single magnitude $v$, since in that case there would be a positive probability of a situation in which it is not clear which response should be considered 'correct', so that $P_{error}$ is not well-defined.) Note that the criterion $\bar{P}_{error}(\theta)$ for ranking encoding rules is not without content, since there exist encoding rules (including DbS) for which the upper bound is less than 1/2 (the error probability in the case of a completely uninformative internal representation).

Let us consider first the case in which there is some part of the diagonal line along which $\tilde{v} = v$ which is not a boundary at which the function $\theta(v, \tilde{v})$ is discontinuous. Then we can choose an open interval $(v_{min}, v_{max})$ such that all values $v, \tilde{v}$ with the property that both $v$ and $\tilde{v}$ lie within the interval $(v_{min}, v_{max})$ are part of a single region on which $\theta(v, \tilde{v})$ is a continuous function. Then let $\theta_{min}$ be the greatest lower bound with the property that $\theta(v, \tilde{v}) \geq \theta_{min}$ for all $v, \tilde{v}$ lying within the specified interval, and similarly let $\theta_{max}$ be the lowest upper bound such that $\theta(v, \tilde{v}) \leq \theta_{max}$ for all values within the specified interval. Because of the continuity of $\theta(v, \tilde{v})$ on this region, as the values $v_{min}, v_{max}$ are chosen to be close enough to each other, the bounds $\theta_{min}, \theta_{max}$ can be made arbitrarily close to one another.

Now for any probabilities $0 \leq \theta \leq \theta' \leq 1$, let $P_{min}(\theta, \theta')$ be the quantity defined in *Equation 30*, when $\theta_1 = \theta$ and $\theta_2 = \theta'$; that is, for any $v_1, v_2$ that are not equal to one another, $P_{min}(\theta, \theta')$ is the probability of an erroneous comparison if the units representing the smaller magnitude each give a 'high' reading with probability $\theta$ and those representing the larger magnitude each give a 'high' reading with probability $\theta'$. Then the probability of erroneous choice $P_{error}$ when $f(v)$ is a distribution with support entirely within the interval $(v_{min}, v_{max})$ is necessarily greater than or equal to the lower bound $P_{min}(\theta_{min}, \theta_{max})$. The reason is that for any $v_1, v_2$ in the support of $f(v)$, the probabilities

$$\theta_i = \int \theta(v_i, \tilde{v}) f(\tilde{v}) d\tilde{v}$$

will necessarily lie within the bounds $\theta_{min} \leq \theta_i \leq \theta_{max}$ for both $i = 1, 2$. Given these bounds, the most

favorable case for accurate discrimination between the two magnitudes will be to assign the largest possible probability $\theta_{max}$ to units being on in the representation of the larger magnitude, and the smallest possible probability $\theta_{min}$ to units being on in the representation of the smaller magnitude. Since the lower bound $P_{min}(\theta_{min}, \theta_{max})$ applies in the case of any individual values $v_1, v_2$ drawn from the support of $f(v)$, this same quantity is also a lower bound for the average error rate integrating over the prior distributions for $v_1$ and $v_2$.

One can also show that as the two bounds $\theta_{min}, \theta_{max}$ approach one another, the lower bound $P_{min}(\theta_{min}, \theta_{max})$ approaches 1/2, regardless of the common value that $\theta_{min}$ and $\theta_{max}$ both approach. Hence it is possible to make $P_{min}(\theta_{min}, \theta_{max})$ arbitrarily close to 1/2, by choosing values for $v_{min}, v_{max}$ that are close enough to one another. It follows that for any bound $P_{min}$ less than 1/2 (including values arbitrarily close to 1/2), we can choose a prior distribution $f(v)$ for which $P_{error}$ is necessarily equal to $P_{min}$ or larger. It follows that in the case of a function $\theta(v, \tilde{v})$ of this kind, the upper bound $\bar{P}_{error}(\theta)$ is equal to 1/2.

In order to achieve an upper bound lower than 1/2, then, we must choose a function $\theta(v, \tilde{v})$ that is discontinuous along the entire line $v = \tilde{v}$. For any such function, let us consider a value $v^*$ with the property that all points $(v, \tilde{v})$ near $(v^*, v^*)$ with $v > \tilde{v}$ belong to one region on which $\theta$ is continuous, and all points near $(v^*, v^*)$ with $v < \tilde{v}$ belong to another region. Then under the assumption of piecewise continuity, $\theta(v, \tilde{v})$ must approach some value $\bar{\theta}(v^*)$ as the values $(v, \tilde{v})$ converge to $(v^*, v^*)$ from within the region where $v > \tilde{v}$, and similarly $\theta(v, \tilde{v})$ must approach some value $\underline{\theta}(v^*)$ as the values $(v, \tilde{v})$ converge to $(v^*, v^*)$ from within the region where $v < \tilde{v}$.

It must also be possible to choose values $v_{min} < v^* < v_{max}$ such that all points $(v, v)$ with $v_{min} < v < v_{max}$ are points on the boundary between the two regions on which $\theta$ is continuous. Given such values, we can then define bounds $\underline{\theta}_{min}$, and $\bar{\theta}_{max}$, such that

$$\underline{\theta}_{min} \leq \theta(v, \tilde{v}) \leq \underline{\theta}_{max}$$

for all $v_{min} < v < \tilde{v} < v_{max}$, and

$$\bar{\theta}_{min} \leq \theta(v, \tilde{v}) \leq \bar{\theta}_{max}$$

for all $v_{min} < \tilde{v} < v < v_{max}$. Moreover, piecewise continuity of the function $\theta(v, \tilde{v})$ implies that by choosing both $v_{min}$ and $v_{max}$ close enough to $v^*$ we can make the bounds $\underline{\theta}_{min}, \underline{\theta}_{max}$ arbitrarily close to $\underline{\theta}(v^*)$, and make the bounds $\bar{\theta}_{min}, \bar{\theta}_{max}$ arbitrarily close to $\bar{\theta}(v^*)$.

Next, for any set of four probabilities $0 \leq \underline{\theta} \leq \underline{\theta}' \leq 1$ and $0 \leq \bar{\theta} \leq \bar{\theta}' \leq 1$, let us define

$$\hat{P}_{min}(\underline{\theta}, \underline{\theta}'; \bar{\theta}, \bar{\theta}') \equiv \mathrm{E}[P_{min}(\theta(z_1), \theta'(z_2)) | z_1 < z_2], \qquad (82)$$

where

$$\theta(z) \equiv z\bar{\theta} + (1-z)\underline{\theta}, \quad \theta'(z) \equiv z\bar{\theta}' + (1-z)\underline{\theta}', \qquad (83)$$

and $z_1, z_2$ are two independent random variables, each distributed uniformly on [0, 1]. Then if $\theta(v, \tilde{v})$ lies between the lower bound $\underline{\theta}$ and upper bound $\underline{\theta}'$ whenever $v < \tilde{v}$, and between the lower bound $\bar{\theta}$ and upper bound $\bar{\theta}'$ whenever $v > \tilde{v}$, then the probability $\theta$ of a processing unit representing the magnitude $v$ giving a 'high' reading will lie between the bounds $\theta(z) \leq \theta \leq \theta'(z)$, where $z = F(v)$ is the quantile of $v$ within the prior distribution. It follows that in the case of any two magnitudes $v_1, v_2$ with $v_1 < v_2$, the probability of an erroneous comparison will be bounded below by $P_{min}(\theta(z_1), \theta'(z_2))$, where $z_i = F(v_i)$ for $i = 1, 2$, since the probability of a correct discrimination will be maximized by making the units representing $v_1$ give as few high readings as possible and the units representing $v_2$ give as many high readings as possible. Integrating over all possible draws of $v_1, v_2$, one finds that the quantity $\hat{P}_{min}(\underline{\theta}, \underline{\theta}'; \bar{\theta}, \bar{\theta}')$ defined in *Equation 82* is a lower bound for the overall probability of an erroneous comparison, given that regardless of the prior $f(v)$, the quantiles $z_1, z_2$ will be two independent draws from the uniform distribution on $[0, 1]$.

Now consider again an encoding function $\theta(v, \tilde{v})$ of the kind discussed two paragraphs above, and an interval of stimulus values $(v_{min}, v_{max})$ of the kind discussed there. For any prior distribution $f(v)$ with support entirely contained within the interval $(v_{min}, v_{max})$, the probability of an erroneous comparison is bounded below by

$$P_n(\theta;f) \geq \hat{P}_{min}(\underline{\theta}_{min}, \underline{\theta}_{max}; \bar{\theta}_{min}, \bar{\theta}_{max}),$$

where the function $\hat{P}_{min}$ is defined in *Equation 82*. Moreover, by choosing the values $v_{min}, v_{max}$ close enough to $v^*$, we can make this lower bound arbitrarily close to $P^e(\underline{\theta}(v^*), \bar{\theta}(v^*))$, where for any probabilities $\underline{\theta}, \bar{\theta}$ we define

$$P^e(\underline{\theta}, \bar{\theta}) \equiv \hat{P}_{min}(\underline{\theta}, \underline{\theta}; \bar{\theta}, \bar{\theta}). \tag{84}$$

Hence in the case of the encoding function considered, the upper bound $\bar{P}_{error}(\theta)$ must be at least as large as $P^e(\underline{\theta}(v^*), \bar{\theta}(v^*))$. We further observe that the quantity $P^e(\underline{\theta}, \bar{\theta})$ defined in *Equation 84* is just the probability of an erroneous comparison in the case of an encoding rule according to which

$$\theta(v, \tilde{v}) = \underline{\theta} \quad \text{if } v < \tilde{v},$$

$$\theta(v, \tilde{v}) = \bar{\theta} \quad \text{if } v > \tilde{v}.$$

Note that in the case of such an encoding rule, the probability of an erroneous comparison is the same for all prior distributions, since under this rule all that matters is the distribution of the quantile ranks of $v$ and $\tilde{v}$. It is moreover clear that $P^e(\underline{\theta}, \bar{\theta})$ is an increasing function of $\underline{\theta}$ and a decreasing function of $\bar{\theta}$. It thus achieves its minimum possible value if and only if $\underline{\theta} = 0$ and $\bar{\theta} = 1$, in which case it takes the value $P_{error}^{DbS}$, the probability of erroneous comparison in the case of decision by sampling (again, independent of the prior distribution).

Thus in the case that there exists any magnitude $v^*$ for which $\underline{\theta}(v^*) > 0$, $\bar{\theta}(v^*) < 1$ or both, there exist priors $f(v)$ for which $P_n(\theta;f)$ must exceed $P_{error}^{DbS} = P^e(0, 1)$. Hence in order to minimize the upper bound $\bar{P}_{error}(\theta)$, it must be the case that $\underline{\theta}(v) = 0$ and $\bar{\theta}(v) = 1$ for all $v$. But then our assumption that the encoding rule $\theta(v, \tilde{v})$ is at least weakly increasing in $v$ and at least weakly decreasing in $\tilde{v}$ requires that

$$\theta(v, \tilde{v}) = 0 \quad \text{for all } v < \tilde{v},$$

$$\theta(v, \tilde{v}) = 1 \quad \text{for all } v > \tilde{v}.$$

Thus the encoding rule must be the DbS rule, the unique rule for which $\bar{P}_{error}(\theta)$ is no greater than $P_{error}^{DbS}$.

## Appendix 7

### Sufficient conditions for the optimality of DbS

Here we consider the general problem of choosing a value of $m$ (the number of samples from the contextual distribution $f(v)$ to use in encoding any individual stimulus) and an encoding rule $\theta(v; \tilde{v}_1, \ldots, \tilde{v}_m)$ to be used by each of the $n$ processing units that encode the magnitude of that single stimulus, so as to minimize the compound objective

$$\bar{P}_{error}(\theta) + K(m),$$

where $\bar{P}_{error}$ is the upper bound on the probability of an erroneous comparison under the encoding rule $\theta$, and $K(m)$ is the cost of using a sample of size $m$ when encoding each stimulus magnitude. The value of $n$ is taken as fixed at some finite value. (This too can be optimized subject to some cost of additional processing units, but we omit formal analysis of this problem.) We assume that $K(m)$ is an increasing function of $m$, and can without loss of generality assume the normalization $K(0) = 0$. In this optimization problem, we assume that the only encoding functions $\theta$ to be considered are ones that are piecewise continuous, at least weakly increasing in $v$, and weakly decreasing in each of the $\tilde{v}_j$.

For any value of $m$, let $P^*(m)$ be the minimum achievable value for $\bar{P}_{error}(\theta)$. (Appendix 6 illustrates how this kind of problem can be solved, for the case $m = 1$.) Then the optimal value of $m$ will be the one that minimizes $P^*(m) + K(m)$.

We can establish a lower bound for $P^*(m)$ that holds for any $m$:

$$
\begin{aligned}
P^*(m) &\equiv \inf_{\theta(v;\tilde{v}_1,\ldots,\tilde{v}_m)} \sup_{f\in\mathcal{F}} P_n(\theta;f) \\
&\geq \sup_{f\in\mathcal{F}} \inf_{\theta(v;\tilde{v}_1,\ldots,\tilde{v}_m)} P_n(\theta;f) \\
&= \sup_{f\in\mathcal{F}} \inf_{\theta(v)} P_n(\theta;f) \equiv \underline{P}_n.
\end{aligned}
\tag{85}
$$

In the second line, we allow the function $\theta(v; \tilde{v}_1, \ldots, \tilde{v}_m)$ to be chosen after a particular prior $f(v)$ has already been selected, which cannot increase the worst-case error probability. In the third line, we note that the only thing that matters about the encoding function chosen in the second line is the mean value of $\theta(v; \tilde{v}_1, \ldots, \tilde{v}_m)$ for each possible magnitude $v$, integrating over the possible samples of size $m$ that may be drawn from the specified prior; hence we can more simply write the problem on the second line as one involving a direct choice of a function $\theta(v)$, which may be different depending on the prior $f(v)$ that has been chosen. The problem on the third line defines a bound $\underline{P}_n$ that does not depend on $m$.

A set of sufficient conditions for $m = 1$ to be optimal is then given by the assumptions that

a. $P^*(0) > P^*(1) + K(1)$, and
b. $P^*(1) - \underline{P} < K(2) - K(1)$.

Condition (a) implies that $m = 0$ will be inferior to $m = 1$: the cost of a single sample is not so large as to outweigh the reduction in $\bar{P}_{error}(\theta)$ that can be achieved using even one sample. Condition (b) implies that $m = 1$ will be superior to any $m' > 1$. The lower bound (*Equation 85*), together with our monotonicity assumption regarding $K(m)$, implies that for any $m' > 1$,

$$P^*(1) - P^*(m') \leq P^*(1) - \underline{P} < K(2) - K(1) \leq K(m') - K(1),$$

and hence that

$$P^*(1) + K(1) < P^*(m') + K(m').$$

While condition (b) is stronger than is needed for this conclusion, the sufficient conditions stated in the previous paragraph have the advantage that we need only consider optimal encoding rules for the cases $m = 0$ and $m = 1$, and the efficient coding problem stated in definition (*Equation 85*), in order to verify that the conditions are both satisfied. The efficient coding problem for the case $m = 1$ is treated in Appendix 6, where we show that $P^*(1) = P_{error}^{DbS} < 1/2$. Using the calculations explained in Appendix 2, we can provide an analytical approximation to this quantity in the limiting case of large $n$.

*Equation 37* states that for any encoding rule $\theta(v)$ and any prior distribution $f(v)$, the value of $P_{error}$ for any large enough value of $n$ will approximately equal

$$P_n(\theta;f) \approx \frac{2}{\sqrt{n\pi}} \int \hat{f}(\tilde{\theta})^2 \sqrt{\tilde{\theta}(1-\tilde{\theta})}\, d\tilde{\theta}, \qquad \text{(37 revisited)}$$

where $\hat{f}(\theta)$ is the probability density function of the distribution of values for $\theta(v)$ implied by the function $\theta(v)$ and the distribution $f(v)$ of values for $v$. In the case of DbS, the probability distribution over alternative internal representations $k_i$ (and hence the probability of error) is the same as in the case of an encoding rule $\theta(v) = F(v)$, so that *Equation 37* can be applied. Furthermore, for any prior distribution $f(v)$, the probability distribution of values for the quantile $z = F(v)$ will be a uniform distribution over the interval $[0, 1]$, so that $\hat{f}(\theta) = 1$ for all $\theta$. It follows that

$$P_{error}^{\text{DbS,lim}} \approx \frac{2}{\sqrt{n\pi}} \int \sqrt{\tilde{\theta}(1-\tilde{\theta})}\, d\tilde{\theta} = \frac{1}{4}\sqrt{\frac{\pi}{n}}. \qquad (86)$$

In the case that $m = 0$, instead, the same function $\theta(v)$ must be used regardless of the contextual distribution $f(v)$. Under the assumption that $\theta(v)$ is piecewise continuous, there must exist a magnitude $v^*$ such that $\theta(v)$ is continuous over some interval $(v_{min}, v_{max})$ containing $v^*$ in its interior. Let $\theta_{min}, \theta_{max}$ be the greatest lower bound and least upper bound respectively, such that

$$\theta_{min} \leq \theta(v) \leq \theta_{max}$$

for all $v_{min} < v < v_{max}$. The continuity of $\theta(v)$ on this interval means that by choosing both $v_{min}$ and $v_{max}$ close enough to $v^*$, we can make both $\theta_{min}$ and $\theta_{max}$ arbitrarily close to $\theta(v^*)$.

By the same argument as in Appendix 6, for any prior distribution $f(v)$ with support entirely contained in the interval $(v_{min}, v_{max})$, the pair of stimulus magnitudes $v_1, v_2$ will have to imply $\theta_{min} \leq \theta(v_1), \theta(v_2) \leq \theta_{max}$ with probability 1, and as a consequence the error probability $P_n(\theta;f)$ will necessarily be greater than or equal to the lower bound $P_{min}(\theta_{min}, \theta_{max})$. By choosing both $v_{min}$ and $v_{max}$ close enough to $v^*$, we can make this lower bound arbitrarily close to $P_{min}(\theta(v^*), \theta(v^*)) = 1/2$. Hence for any encoding rule $\theta(v)$ with $m = 0$, the upper bound $\bar{P}_{error}(\theta)$ cannot be lower than $1/2$. It follows that $P^*(0) = 1/2$.

Given this, condition (a) can alternatively be expressed as

$$P_{error}^{DbS} + K(1) < 1/2.$$

Note that if $K(1)$ remains less than $1/2$ no matter how large $n$ is, this condition will necessarily be satisfied for all large enough values of $n$, since *Equation 86* implies that $P_{error}^{DbS}$ eventually becomes arbitrarily small, in the case of large enough $n$. (On the other hand, the condition can easily be satisfied for some range of smaller values of $n$, even if $K(1) > 1/2$ once $n$ becomes very large.)

In order to consider the conditions under which condition (b) will also be satisfied, it is necessary to further analyze the efficient coding problem stated in *Equation 85*. We first observe that for any prior $f(v) \in \mathcal{F}$ and encoding rule $\theta(v)$, the encoding rule can always be expressed in the form $\theta(v) = \varphi(F(v))$, where $\varphi(z)$ is a piecewise-continuous, weakly increasing function giving the probability of a 'high' reading as a function of the quantile $z$ of the stimulus magnitude in the prior distribution. We then note that when this representation is used for the encoding function in *Equation 85*, the error probability $P_n(\theta;f)$ depends only on the function $\varphi(z)$, in a way that is independent of the prior $f(v)$. Hence the inner minimization problem in *Equation 85* can equivalently be written as

$$\inf_{\varphi(z)} P_n(\varphi). \qquad (87)$$

This problem has a solution for the optimal $\varphi(z)$ for any number of processing units $n$, and an associated value that is independent of the prior $f(v)$. Hence we can write the bound defined in *Equation 85* more simply as

$$\underline{P}_n = \inf_{\varphi(z)} P_n(\varphi). \qquad (88)$$

Condition (b) will be satisfied as long as the bound defined in *Equation 88* is not too much lower

than $P_{\text{error}}^{\text{DbS}}$. In fact, this bound can be a relatively large fraction of $P_{\text{error}}^{\text{DbS}}$. We consider the problem of the optimal choice of an encoding function $\theta(v)$ for a known prior $f(v)$ in Appendix 2. In the limiting case of a sufficiently large $n$, substitution of *Equation 2* into 37 yields the approximate solution

$$\underline{P}_n^{\text{lim}} \approx \frac{2}{\sqrt{n\pi}} \frac{1}{\pi^2} \frac{d\tilde{\theta}}{\sqrt{\tilde{\theta}(1-\tilde{\theta})}} = \frac{2}{\sqrt{n\pi^3}}. \tag{89}$$

Thus as $n$ is made large, the ratio $\underline{P}_n^{\text{lim}}/P_{\text{error}}^{\text{DbS,lim}}$ converges to the value

$$\underline{P}^{\text{lim}}/P_{\text{error}}^{\text{DbS,lim}} = 8/\pi^2 = 0.81. \tag{90}$$

This means that increases in the sample size $m$ above one cannot reduce $P^*(m)$ by even 20 percent relative to $P^*(1)$, no matter how large the sample may be, whereas $P^*(1)$ may be only a small fraction of $P^*(0)$ (as is necessarily the case when $n$ is large). This makes it quite possible for $K(2) - K(1)$ to be larger than $P_{\text{error}}^{\text{DbS}} - \underline{P}$ while at the same time $P^*(0) - P_{\text{error}}^{\text{DbS}}$ is larger than $K(1)$. **In this case, the optimal sample size will be $m = 1$, and the optimal encoding rule will be DbS**.

While these analytical results for the asymptotic (large-$n$) case are useful, we can also numerically estimate the size of the terms $P^*(0), \underline{P}$, and $P_{\text{error}}^{\text{DbS}}$ in the case of any finite value for $n$. We have derived an exact analytical value for $P^*(0) = 1/2$ above. The quantity $P_{\text{error}}^{\text{DbS}}$ can be computed through Monte Carlo simulation for any value of $n$. (Note that this calculation depends only on $n$, and is independent of the contextual distribution $f(v)$; we need only to calculate $P_n(\varphi)$ for the function $\varphi(z) = z$.) The calculation of $\underline{P}_n$ for a given finite value of $n$ is instead more complex, since it requires us to optimize $P_n(\varphi)$ over the entire class of possible functions $\varphi(z)$.

Our approach is to estimate the minimum achievable value of $P_n(\varphi)$ by finding the minimum achievable value over a flexible parametric family of possible functions $\varphi(z)$. We specify the function $\varphi$ in terms of the implied $\hat{F}(\theta)$, the CDF for values of $\theta(v)$. We let $\hat{F}(\theta)$ be implicitly defined by

$$[\sin((\pi/2)\hat{F}(\theta))]^2 = g(\theta), \tag{91}$$

where $g(\theta)$ is a function of $\theta$ with the properties that $g(0) = 0$, as required for $\hat{F}(\theta)$ to be the CDF of a probability distribution. More specifically, we assume that $g(\theta)$ is a finite-order polynomial function consistent with these properties, which require that it can be written in the form

$$g(\theta) = \theta\big[1 + (\theta - 1)\big(g_0 + g_1\theta + \ldots + g_p\theta^p\big)\big], \tag{92}$$

where $\{g_0, \ldots, g_p\}$ are a set of parameters over which we optimize. Note that for a large enough value of $p$, any smooth function can be well approximated by a member of this family. At the same time, our choice of a parametric family of functions has the virtue that the CDF that corresponds to the optimal coding rule in the large-$n$ limit belongs to this family (regardless of the value of $p$), since this coding rule (*Equation 3*) corresponds to the case $g_0 = \ldots = g_p = 0$ of *Equation 92*.

We computed via numerical simulations the best encoder function assuming $g(\theta)$ to be of order 5 (*Equation 92*) for various finite values of $n = [5, 10, 15, 20, 25, 30, 35, 40]$, and we define the expected error of this optimal encoder for a given $n$ to be $\underline{P}_n^g$ (i.e., a lower bound for $P_n$ within the family of functions defined by $g$). Our goal is to compare this quantity to the asymptotic approximation $\underline{P}_n^{\text{lim}}$, in order to evaluate how accurate the asymptotic approximation is.

Additionally, we also compute the value $P_{\text{error}}^{\text{DbS}}$ for each finite value of $n$ through Monte Carlo simulation (please note that $P_{\text{error}}^{\text{DbS}}$ is different from the quantity $P_{\text{error}}^{\text{DbS,lim}}$ defined in *Equation 86*, that is only an asymptotic approximation for large $n$). Then, we can compare $P_{\text{error}}^{\text{DbS}}$ to the value predicted by the asymptotic approximations $P_{\text{error}}^{\text{DbS,lim}}$ and $\underline{P}_n^{\text{lim}}$.

Another quantity that is important to compute, in order to determine whether DbS can be optimal when $n$ is not too large, is the size of $P^*(0)$ relative to the quantities computed above. Since $P^*(0)$ does not shrink as $n$ increases, it is obvious that $P^*(0)$ is much larger than the other quantities in the large-$n$ limit. But how much bigger is it when $n$ is small? To investigate this, we compute the value of the ratio $P^*(0)/\underline{P}_n^{\text{lim}}$ when $n$ is small. This quantity is given by

$$\frac{P^*(0)}{\underline{P}_n^{\lim}} = \frac{\sqrt{n\pi^3}}{4} \tag{93}$$

In *Appendix 7—figure 1*, all error quantities discussed above are normalized relative to $\underline{P}_n^{\lim}$. The black dashed lines in both panels represent $(\underline{P}_n^{\lim}/\underline{P}_n^{\lim}) = 1$. The ratio of the asymptotic approximation for $P_{\text{error}}^{\text{DbS,lim}}$ relative to $\underline{P}_n^{\lim}$ is plotted with the red dashed lines, where $(P_{\text{error}}^{\text{DbS,lim}}/\underline{P}_n^{\lim}) \approx 1.23$. Note that the sufficient conditions for DbS to be optimal can be stated as

a. $K(1) < P^*(0) - P_{\text{error}}^{\text{DbS}}$, and
b. $K(2) - K(1) > P_{\text{error}}^{\text{DbS}} - \underline{P}_n$.

Therefore, *Appendix 7—figure 1* shows the numerical magnitudes of the expressions on the right-hand side of both inequalities (normalized by the value of $\underline{P}_n^{lim}$). The most important result from the analyses presented in this figure is that even for small values of , the right-hand side of the first inequality (see right panel) will be a much larger quantity than the right-hand side of the second inequality (see left panel). Thus it can easily be the case that $K(1)$ and $K(2)$ are such that both inequalities are satisfied: it is worth increasing $m$ from 0 to 1, but not worth increasing $m$ to any value higher than 1. In this case, the optimal sample size will be $m = 1$, and the optimal encoding rule will be DbS.

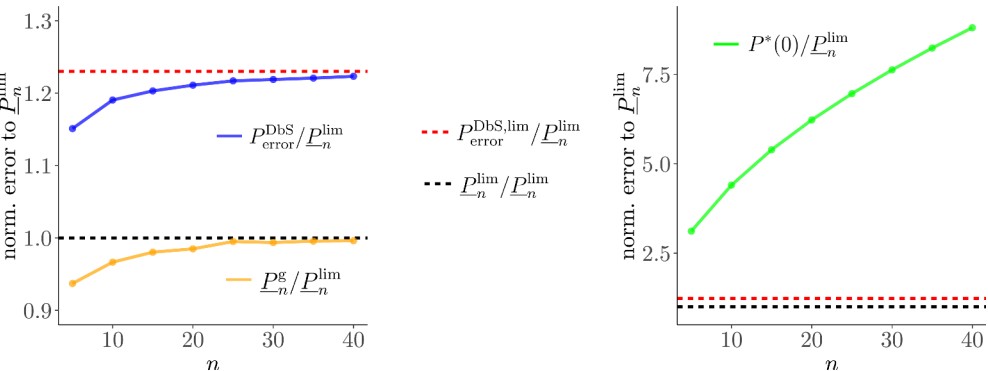

**Appendix 7—figure 1.** Performance of efficient coding rules.

Additionally, we found that the computations of $P_{\text{error}}^{\text{DbS}}$ for each finite value of $n$ are slightly higher than $\underline{P}_n^{\lim}$ even for small $n$ values (blue line in the left panel), but quickly reach the asymptotic value $P_{\text{error}}^{\text{DbS,lim}}/\underline{P}_n^{\lim}$ as $n$ increases. Thus, even for small values of $n$, the asymptotic approximation of optimal performance for the case of complete prior knowledge is superior than DbS. We also found that the computations of $\underline{P}_n^{\text{g}}$ for each finite value of $n$ cannot reduce $\underline{P}_n^{\lim}$ by even five percent for small $n$ values (orange line in the left panel). Moreover, $\underline{P}_n^{\text{g}}$ quickly reached the asymptotic value $\underline{P}_n^{\lim}$, thus suggesting that the asymptotic solution is virtually indistinguishable from the optimal solution (at least based on the flexible family of $g$ functions) also for finite values of $n$, which crucially are in the range of the values found to explain the data in the numerosity discrimination experiment of our study. Thus, these results confirm that the asymptotic approximations used in our study are not likely to influence the conclusions of the experimental data in our work.

## Appendix 8

### Relation to *Bhui and Gershman, 2018*

*Bhui and Gershman, 2018* also argue that an efficient coding scheme can be implemented by a version of DbS. However, both the efficient coding problem that they consider, and the version of DbS that they consider, are different than in our analysis, so that our results are not implied by theirs.

Like us, Bhui and Gershman consider encoding schemes in which the internal representation $r$ must take one of a finite number of values. However, their efficient coding problem considers the class of all encoding rules that assign one or another of $N$ possible values of $r$ to a given stimulus $v$. In their discussion of the ideal efficient coding benchmark, they do not require $r$ to be the ensemble of output states of a set of $n$ neurons, each of which must use the same rule as the other units, and therefore consider a more flexible family of possible encoding rules, as we explain in more detail below.

The encoding rule that solves our efficient coding problem is stochastic; even under the assumption that the prior $f(v)$ is known with perfect precision (the case of unbounded $m$ in the more general specification of our framework, so that sampling error in estimation of this distribution from prior experience is not an issue), we show that it is optimal for the probabilities $p(k|v)$ not to all equal either zero or one. The optimal rule within the more flexible class considered by Bhui and Gershman is instead deterministic: each stimulus magnitude $v$ is assigned to exactly one category $k$ with certainty. The boundaries between the set of $n + 1$ categories furthermore correspond to the quantiles $(1/(n + 1), 2/(n + 1), \ldots, n/(n + 1))$ of the prior distribution, so that each category is used with equal frequency. Thus the optimal encoding rule is given by a deterministic function $y(v)$, a non-decreasing step function that takes $n + 1$ discrete values.

Bhui and Gershman show that when there is no bound on $m$, the number of samples from prior experience that can be used to estimate the contextual distribution — their optimal encoding rule for a given number of categories $N$ — can be implemented by a form of DbS. However, the DbS algorithm that they describe is different than in our discussion. Bhui and Gershman propose to implement the deterministic classification $y(v)$ by computing the fraction of the sampled values $\tilde{v}$ that are less than $v$. In the limiting case of an infinite sample from the prior distribution, this fraction is equal to $F(v)$ with probability one, and $y(v)$ is then determined by which of the intervals $[0, 1/N], [1/N, 2/N], \ldots, [(N-1)/N, 1]$ the quantile $F(v)$ falls within. Thus whereas in our discussion, DbS is an algorithm that allows each of our units to compute its state using only a single sampled value $\tilde{v}_j$, the DbS algorithm proposed by Bhui and Gershman to implement efficient coding is one in which a large number of sampled values are used to jointly compute the output states of all of the units in a coordinated way.

Bhui and Gershman also consider the case in which only a finite number of samples $(\tilde{v}_1, \ldots, \tilde{v}_m)$ can be used to compute the representation $k_i$ of a given stimulus magnitude $v_i$, and ask what kind of rule is efficient in that case. They show that in this case a variant of DbS with kernel-smoothing is superior to the version based on the empirical quantile of $v_i$ (which now involves sampling error). In this more general case, the variant DbS algorithms considered by Bhui and Gershman make the representation $k_i$ of a given stimulus probabilistic; but the class of probabilistic algorithms that they consider remains different from the one that we discuss. In particular, they continue to consider algorithms in which the category $k_i$ can be an arbitrary function of $v_i$ and a single set of $m$ sampled values that is used to compute the complete representation; they do not impose the restriction that $k_i$ be the number of units giving a 'high' reading when the output state of each of $n$ individual processing units is computed independently using the same rule (but an independent sample of values from prior experience in the case of each unit).

The kernel-smoothing algorithms that they consider are based on a finite set of $m$ pairwise comparisons between the stimulus magnitude $v_i$ and particular sampled values $\tilde{v}_j$, the outcomes of which are then aggregated to obtain the internal representation $k_i$. However, they allow the quantity $K(v_i - \tilde{v}_j)$ computed by comparing $v_i$ to an individual sampled value to vary continuously between 0 and 1, rather than having to equal either 0 or 1, as in our case (where the state of an individual unit must be either 'high' or 'low'). The quantities $K(v_i - \tilde{v}_j)$ are able to be summed with perfect precision, before the resulting sum is then discretized to produce a final representation that takes one of only $N$ possible values. Thus an assumption that only finite-precision calculations are possible is

made only at the stage where the final output of the joint computation of the processors must be 'read out'; the results of the individual binary comparisons are assumed to be integrated with infinite precision. In this respect, the algorithms considered by Bhui and Gershman are not required to economize on processing resources in the same sense as the class that we consider; the efficient coding problem for which they present results is correspondingly different from the problem that we discuss for the case in which $m$ is finite.

