## [Decision Letter]

**Acceptance summary:**

We believe that this paper makes fundamental contributions to our understanding of the perception-decision interface. In particular, it sheds light on how perceptual representation adapts to task demands. We foresee that this paper will stimulate future experimental work to test the mechanistic hypotheses postulated by the theory.

**Decision letter after peer review:**

Thank you for submitting your article "Efficient sampling and noisy decisions" for consideration by *eLife*. Your article has been reviewed by three peer reviewers, including Samuel J Gershman as the Reviewing Editor and Reviewer #1, and the evaluation has been overseen by Joshua Gold as the Senior Editor. The following individuals involved in review of your submission have agreed to reveal their identity: Konstantinos Tsetsos (Reviewer #2); Sebastian Gluth (Reviewer #3).

The reviewers have discussed the reviews with one another and the Reviewing Editor has drafted this decision to help you prepare a revised submission.

Summary:

In their manuscript, Heng and colleagues derive general optimality principles and predictions of a decision-making system (or agent) that is restricted to encode information in terms of a finite number of binary samples. They find that in case of maximizing accuracy, the principle aligns with mutual information, but this changes if reward maximization is taken into account. Based on this, they conduct three numerosity experiments with different incentive rules ( "perceptual" vs. "preferential"). Surprisingly, they find that participants do not adjust their behavior to the incentive rule. Instead their behavior is best explained by the Decision by Sampling (DbS) framework, which assumes that behavior depends on the distribution of stimuli in the environment (in contrast to a simple log rule), but not in an optimal way. The reviewers agreed that this is an interesting and potentially important contribution to the literature on judgment and decision making, but requires substantive revisions in a number of respects detailed below.

Essential revisions:

1) Technical/conceptual issues.

a) The theoretical results depend on asymptotically taking n to be large. When n is finite, the proposed coding schemes may not necessarily be optimal, which should be recognized. Indeed this forms the basis of the Bhui and Gershman analysis of DbS.

b) The authors write that DbS continues to "explain the shape of ubiquitous psycho-economic functions", but they also provide accuracy maximization results. Do the alternative optimization criteria affect these curves or not?

c) The reward maximizing scheme assumes that agents seek to minimize "regret" (v1-v2). This is a viable hypothesis, but how does optimal encoding change if instead agents care just about the obtained reward (say, v1)? More generally, it is not a given that relative and not absolute reward is the relevant quantity in value based tasks.

2) Experimental/analysis issues.

a) The model recovery results shown in Figure 3D as well as the fits in Figure 4 rely on models in which the shape of the prior distribution is fixed and equal to the shape of the prior distribution used in experiments 1-2. Is the encoding rule still identifiable if the parameter controlling the shape of the prior is free to vary? The authors show recovery of the α parameter within the DbS model (Figure 5) but not when the encoding rule is unknown. Crucially, the conclusion that DbS outperform the other two encoding rules can be undermined if letting the prior free to vary induces model mimicry. Please examine this possibility. If indeed the different encoding schemes are not falsifiable it should be clearly stated that the conclusions (e.g "we found that humans employ a less optima strategy.", "allowed us to test the hypothesis" etc) hold under the specific assumption that the prior distribution is fixed.

b) Modeling of the adaptation to priors in experiments 1-3 assumes by definition that the prior parameter starts from a higher than the nominal value, and adapts across time with a time-scale that is shared across experiments. Is there indeed need/ evidence for adaptation? Observing the data in Figure 4—figure supplement 2 I can see that the accuracy a) is stable across time-rendering any adaptation process counterintuitive and, b) accuracy in Experiments 1-2 is higher than the accuracy in Experiment 3. Thus, the lower asymptotic α appears to serve the role of lowering overall accuracy. Please 1) superimpose the across time accuracy of the DbS model with prior adaptation on the traces shown in Figure 4—figure supplement 2, in order to see if the model systematically misfits the data by starting with α=2.84. 2) Please compare the fits of the adapting prior model with the a) the fits of a DbS model with just a free α parameter and b) a DbS model with α=2 and n as free parameter. Can these alternatives explain the data more parsimoniously?

c) One possible reason why the two conditions did not lead to differences could be that – after doing one condition for two days – it might have been impossible for the participants to adjust their "habit-like" behavior to a new incentive rule. This could be checked by analyzing the first half of the task in a between-subject manner.

3) Expository issues.

a) Provide more intuition for the equivalence between results under different optimization criteria. Do any of these results rely on the asymptotics?

b) The clarity of the Introduction can be improved. In the second paragraph, the authors suddenly jump to a discussion of differences between perceptual and preferential choice, but I think it would be more important to first make clear what the overall goal of the work is. The third paragraph is very confusing. Its first sentence is not even a full sentence (a verb is missing at "where only a finite number.…") and pretty much incomprehensible. Then, the work of Simon Laughlin is discussed, but it is questionable whether this is really the best way to motivate the proposal of a binary encoding system (why not simply saying that neurons provide binary outcomes). The reference to Query Theory in the fourth paragraph remains vague. In a later paragraph, the idea of adaptation to a frequency distribution is introduced without explaining what it actually means (and one reason for this is that DbS is not well explained in the previous paragraph).

c) In the Abstract, the authors should make more clear what the task was about (though we understand that this isn't easy give the word limit). In addition, the word "Here" is used to start two consecutive sentences, and an "a" is missing at "strategy that might be utilized…".

4) Links to related literature.

a) Do the results cast doubt on the argument made in the paper by Rustichini et al., 2017, which argues for a coding scheme based on expected utility maximization rather than mutual information?

b) Clarify that Equation 9 only corresponds to DbS in the asymptotic limit. The finite sample regime was emphasized in Bhui and Gershman, 2018, in order to explain certain phenomena (such as range effects) that do not follow directly from the CDF encoding function. Instead, that paper showed how these results could be obtained from a smoothed encoding function computed on a small set of samples. Relatedly, please clarify the links to the Bhui and Gershman paper. In particular, how does infomax in this paper (like in Supplementary Note 1) related to infomax in their paper? Also their work is described as adding noise after efficient coding, but this is not the case. The "noise" in that model comes purely from the fact that a finite number of samples are drawn, so that the sample-based CDF only approximates the true CDF.

c) The authors should discuss (and scrutinize) their empirical findings a bit more in the context of other studies that have compared perceptual and preferential decisions, in particular Dutilh and Rieskamp, 2015, who studied a quite similar task (choosing based on the number of dots with a perceptual vs. a preferential incentive rule). Here, it was found that decisions were slowest for the most difficult trials in the perceptual condition but not in the preferential condition. The question is whether there are any interesting, related response time differences between the two conditions in the current task (because if the number of dots on the left and right is very similar, one should not think too long about it if the reward depends on the number of dots [preferential], but one would need to think for a long time about it to decipher which side has more dots [perceptual]).

[Editors' note: further revisions were suggested prior to acceptance, as described below.]

Thank you for resubmitting your article "Efficient sampling and noisy decisions" for consideration by *eLife*. Your revised article has been reviewed by three peer reviewers, including Samuel J Gershman as the Reviewing Editor and Reviewer #1, and the evaluation has been overseen by Joshua Gold as the Senior Editor. The following individuals involved in review of your submission have agreed to reveal their identity: Konstantinos Tsetsos (Reviewer #2); Sebastian Gluth (Reviewer #3).

The reviewers have discussed the reviews with one another and the Reviewing Editor has drafted this decision to help you prepare a revised submission. We are optimistic that the next revision will be acceptable for publication.

Summary:

In their revision the authors have successfully addressed most of the points we had raised. In particular the authors now discuss in detail the role of asymptotics and the relationship between their framework and the one proposed by Bhui and Gershman. This development has resulted in the extension of the framework in order to capture finite sampling from the prior distribution. Additionally, the authors have demonstrated that the encoding rule (as well as the shape of the prior and the number of samples) is identifiable when the shape of the prior is free to vary. All these developments have sufficiently improved the manuscript.

1) Having established the identifiability of the α parameter under the DbS model, it seems imperative to fit Experiment 3 using α as a free parameter and omitting the adaptation mechanism (this has been done in the revision but these results are used to examine whether there is adaptation or not, rather than to actually examine if the fitted prior differs between experiments 1-2 and 3). In other words, Figure 5A can be expanded to include the α fits from Experiment 3. This exercise can address whether there is indeed a change to the shape of the prior across experiments, which is a pivotal component of the proposed framework relative to alternative frameworks that assume complex representational non-linearities without sensitivity to the prior. The results in Figure 5C show that α in Experiment 3 converges to a lower asymptotic value. However, imposing an adaptation process, especially when there is no strong support for such process, can obscure the interpretation of the fits. Furthermore, I remain skeptical about the claim that there is dynamical adaptation within each experiment: i) if anything, the new analyses show that the "free α" model provides a better goodness of fit, and ii) Figures 4—figure supplement 2 and Figure 5—figure supplement 1 show no obvious dynamical trends in behavior. How does the adaptation manifest itself in the data? Figure 5—figure supplement 2 hints toward an early period in which the "free-α" fits worse than the dynamic α model (up to ~150 trials). Perhaps modeling this discrepancy explicitly (e.g. two α parameters for early and late trials, respectively) would suffice.

---

## [Author Response]

Essential revisions:1) Technical/conceptual issues.a) The theoretical results depend on asymptotically taking n to be large. When n is finite, the proposed coding schemes may not necessarily be optimal, which should be recognized. Indeed this forms the basis of the Bhui and Gershman analysis of DbS.

In the response to this point, we will also address other related comments: In point 4b, reviewers ask to clarify the links to relation to Bhui and Gershman and the asymptotic limit statements. In minor point a, the reviewers ask to provide more intuition for the equivalence between results under different optimization criteria, and whether any of these results rely on asymptotics.

We would like to highlight that your comments motivated us to improve the presentation and interpretation of the theoretical results. We have now redefined the elaboration of our theoretical framework which we hope will clarify various issues raised by the reviewers in this comment. This led us to re-structure the presentation of the theoretical results in the revised manuscript. Importantly, none of our conclusions are affected by these reformulations and analyses. Below, we provide a brief description of how our revision addresses the reviewers’ concerns.

1) In the re-elaboration of our framework, we now consider two distinct types of resource constraints and their implications for the nature of an efficient coding scheme. The first cognitive resource that we consider is the number of samples *n* that can be used to encode the magnitude of an individual stimulus, as formulated in the initial submission. However, our framework now formalizes a second resource constraint as well: the number of samples *m* from the prior that can be used to adapt the coding rule to a particular context. For the case of unbounded prior knowledge (i.e., perfect knowledge of the prior, as is typically assumed in the efficient coding frameworks in early sensory systems), we derive analytical solutions for the case when the system maximizes either accuracy or fitness (these results were presented in the initial submission). We now make clear at various points in the text that these results rely on asymptotic approximations. However, we now also analyze numerically how far these asymptotic approximations are from the actual optimal solutions for small values of *n* (crucially, including the range of values needed to fit the experimental data). We found via numerical approximations that the analytical solutions appear to be nearly optimal for the relevant range of *n* in our experiments (see Appendix 7—figure 1). Crucially, for a number *n* of samples in this range, both the numerical-optimal and large-*n* analytical approximation (again under the assumption of full prior knowledge) are found to be more efficient than the DbS model. Therefore, we argue that our conclusions in the initial submission regarding the way in which an optimal encoding rule (in the case of no limit on *m*) is different from DbS still hold for the range of values *n* that are relevant to explain our data.

2) As just discussed, we obtain analytical solutions that approximately characterize the optimal encoding rule *θ*(*v*) in the limit as *n* is made sufficiently large. It should be noted however that we are always assuming that *n* is finite, and that this constrains the accuracy of the decision maker’s judgments, while *m* (knowledge about the prior) is instead assumed to be unbounded in this part of the analysis and hence no constraint.

We then consider instead the case when it is costly for a system to have full knowledge of the prior distribution (because the system must dynamically adapt to different prior distributions in changing contexts). In this case, it can economize on processing resources to use only a few samples from past experiences rather than requiring the system to completely learn the prior distribution. Our general framework allows us to define formally an optimization problem for this case as well. Crucially, *we have formally demonstrated* that under the assumption that reducing either *m* or *n* allows one to economize on scarce cognitive resources, it is under certain circumstances (that we precisely define) most efficient to use an algorithm with a very low value of *m* (as assumed by DbS), while allowing *n* to be much larger. This allows us to state conditions under which DbS will be the optimal encoding rule. Moreover, we wish to emphasize that this result is derived for the case of a particular finite number of processing units *n* (and a corresponding finite total number of samples from the contextual distribution used to encode a given stimulus), and does not assume that n must be large (see revised Results section, and Appendixes 6 and 7). We believe that these results further extend the novel contributions of our work, as they provide a formal explanation of why DbS might better describe behavior in our experiments than the rules that would be optimal for the particular stimulus distributions that we use, and, more generally, show why DbS might be a good strategy to be used by higher-order level systems that need to flexibly adapt to dynamical contexts.

3) Regarding the relation of our work to Bhui and Gershman, we have now dedicated a paragraph to this topic in the Discussion section where we provide a high-level explanation of the similarities and differences between our studies. Moreover, we discuss in more detail the relation and differences between our studies in a new supplementary note (Appdendix 8, in the revised manuscript).

While Bhui and Gershman provide an argument for a version of DbS as the solution to an efficient coding problem, their definition of the efficient coding problem is quite different from ours, so that our result is quite distinct. Moreover, their conclusions differ from ours, given that different classes of feasible algorithms are considered. In Bhui and Gershman, the classic formulation of DbS is found to be optimal only in the limiting case in which a very large number of samples can be used for the encoding; in the case of a smaller sample, they find that a smoothed version of DbS improves upon the classic formulation. In our analysis, instead, the classic formulation of DbS is found to be optimal when the cost of using a greater number of samples is sufficiently important; instead, when it is economical to use a larger number of samples, we show that an alternative coding rule is superior to DbS. While the flavor of these results may seem directly opposed to each other, there is no inconsistency between them, as the classes of encoding rules considered are quite different in the two papers.

Briefly, Bhui and Gershman consider a case in which only a finite number of values can be sampled from the prior, and show that a variant of DbS with kernel-smoothing is superior to its standard version. However, a key difference from our analysis is that they allow the kernel-smoothed quantity (computed by comparing the input *v* with a sample ˜*v* from the prior distribution) to vary continuously between 0 and 1, rather than requiring the output of each processing unit to be either 0 or 1 as in our implementation (Figure 1 in the revised manuscript). When instead we consider only coding schemes that encode information based on zeros or ones, we show that coding efficiency can be improved relative to DbS, but only in the case that is sufficiently cheap to obtain a large sample from the prior distribution. In the case that a large sample from the prior distribution is available, we offer both an analytical solution for the optimal encoding rule in the large-*n* limit, and a numerical solution for it in the case of smaller (empirically realistic) values of *n*, and show that it is possible to improve upon DbS in both cases.

b) The authors write that DbS continues to "explain the shape of ubiquitous psycho-economic functions", but they also provide accuracy maximization results. Do the alternative optimization criteria affect these curves or not?

We apologize for our lack of clarity in this statement. The alternative optimization criteria do affect these curves, as we show in Figure 2B (in the revised manuscript). For all of the optimization objectives that we consider, the encoding rules share the qualitative feature that they are steeper for regions of the colorred stimulus space with higher prior density. However, mild changes in the steepness of the curves will be represented in significant discriminability differences between the different encoding rules across the support of the prior distribution (Figure 2D). We have now clarified this point in the revised manuscript.

c) The reward maximizing scheme assumes that agents seek to minimize "regret" (v1-v2). This is a viable hypothesis, but how does optimal encoding change if instead agents care just about the obtained reward (say, v1)? More generally, it is not a given that relative and not absolute reward is the relevant quantity in value based tasks.

It is not correct that in our discussion of the case of “expected reward maximization”, we assume that the decision maker cares about relative rather than the absolute reward achieved in a given outcome. We acknowledge that this misunderstanding might have been triggered by our description of the problem in the Results section of the initial submission. We have revised the corresponding appendix (Appendix 5 in the revised manuscript) and the Results section to clarify our calculations.

The objective that we maximize in this case is E[*v*(chosen)], the expected value (in absolute rather than relative terms) of the item that is chosen. In any given state (defined by the values (*v*_1_*,v*_2_)), we assume that is only the value of the chosen item that matters for the decision maker’s payoff, not the value of the item that was available but not chosen; our theory is thus very different from a model of “regret.” We now show explicitly why maximization of the objective criterion stated in the initial submission is equivalent to minimizing expected loss (Equation 59, in Appendix 5 of the revised manuscript). The latter formulation is then useful in our subsequent calculations. But these calculations in no way assume a concern with relative valuations. Please see Appendix 5.

2) Experimental/analysis issues.a) The model recovery results shown in Figure 3D as well as the fits in Figure 4 rely on models in which the shape of the prior distribution is fixed and equal to the shape of the prior distribution used in experiments 1-2. Is the encoding rule still identifiable if the parameter controlling the shape of the prior is free to vary? The authors show recovery of the α parameter within the DbS model (Figure 5) but not when the encoding rule is unknown. Crucially, the conclusion that DbS outperform the other two encoding rules can be undermined if letting the prior free to vary induces model mimicry. Please examine this possibility. If indeed the different encoding schemes are not falsifiable it should be clearly stated that the conclusions (e.g "we found that humans employ a less optima strategy.", "allowed us to test the hypothesis" etc) hold under the specific assumption that the prior distribution is fixed.

Thanks for this observation, this is an excellent suggestion to test whether our competing models are falsifiable. Therefore, we carried out the recovery analyses suggested by the reviewers (i.e., leaving both *α* and *n* as free parameters). We found that it is possible to clearly distinguish DbS from the full-prior knowledge optimal models. We note that the distinction between the optimal models (accuracy and fitness) is more difficult for lower values of *α*, but still possible for the values of *α* = 2 assumed in experiments 1 and 2. This *α* ≈ 0 limiting-case behavior is expected based on our normative results. Recall that the shape of the prior distribution *f*(*v*) used in our experiments is given byf(v)=c(1−v)awhich means that as *α* approaches 0, *f*(*v*) approaches a uniform distribution. Also recall that the encoding functions for accuracy and fitness maximization are given byθ(v)=sin[π2∫−∞vf(v~)dv~]andθ(v)=sin[π2•c∫−∞vf(v~)2/3dv~]2respectively. This means that as the prior distribution approaches a uniform distribution, the encoding rules will be indistinguishable. Thus, this provides an explanation of why it is more difficult to distinguish the optimal rules from each other as *α* approaches 0 for our prior distribution. Nevertheless, we emphasize that DbS will always be distinguishable from the optimal encoding rules. In addition to correct model identification, the parameters *α* and *n* were successfully recovered for each generating model. These results thus provide clear evidence that, first, the sampling models are falsifiable, and second, DbS is the most likely strategy used by the participants among the tested models. We present the results of the model recovery analyses in Figure 3—figure supplement 3, (in panels a, b and c, the generating models are Accuracy (blue), Reward (red) and DbS (green), respectively).

b) Modeling of the adaptation to priors in experiments 1-3 assumes by definition that the prior parameter starts from a higher than the nominal value, and adapts across time with a time-scale that is shared across experiments. Is there indeed need/ evidence for adaptation? Observing the data in Figure 4—figure supplement 2 I can see that the accuracy a) is stable across time-rendering any adaptation process counterintuitive and, b) accuracy in Experiments 1-2 is higher than the accuracy in Experiment 3. Thus, the lower asymptotic α appears to serve the role of lowering overall accuracy. Please 1) superimpose the across time accuracy of the DbS model with prior adaptation on the traces shown in Figure 4—figure supplement 2, in order to see if the model systematically misfits the data by starting with α=2.84. 2) Please compare the fits of the adapting prior model with the a) the fits of a DbS model with just a free α parameter and b) a DbS model with α=2 and n as free parameter. Can these alternatives explain the data more parsimoniously?

Regarding the first questions in this point, here we would like to clarify that average performance across experiments should not be compared back to back across experiments given that the set of stimuli presented on each trial were selected to roughly match performance across experiments (based on model simulations) while taking into consideration the different encoding functions for each empirical prior distribution. In any case, we now provide the model predictions of the best model alongside the observed performance (new Figure 5—figure supplement 1). However, we agree with the reviewers that evidence for adaptation should be studied more in detail. As a first approach, we performed out-of-sample model comparisons based on the models proposed by the reviewers: (i) adaptive-*α* model, (ii) model with free *α* but non-adapting over time, and (iii) model with *α* = 2. The results of the out-of-sample predictions revealed that the best model was the free-*α* model, followed closely by the adaptive-*α* model (∆LOO = 1.8) and then by fixed-*α* model (∆LOO = 32.6). However, we did not interpret the apparent small difference between the adaptive-*α* and free-*α* model as evidence for lack of adaptation, given that the more complex adaptive model will be strongly penalized after adaptation is stable. That is, if adaptation is occurring, then the adaptive-*α* only provides a better fit for the trials corresponding to the adaptation period. After adaptation the adaptive-*α* model should provide a similar fit than the free-*α* model, however with a larger complexity that will be penalized by model comparison metrics. Therefore, to investigate the presence of adaptation, we took a closer quantitative look at evolution of the fits across trial experience. We computed the average trial-wise predicted Log-Likelihood (by sampling from the hierarchical Bayesian model) and compared the differences of this metric between the competing models and the adaptive model. We hypothesized that if adaptation is taking place, the adaptive-*α* model would have an advantage relative to the free-*α* model at the beginning of the session, with these differences vanishing toward the end. On the other hand, the fixed-*α* should roughly match the adaptive-*α* model at the beginning and then become worse over time, but these differences should stabilize after the end of the adaptation period. This is exactly what we found (Figure 5—figure supplement 2). These results thus provide evidence of adaptation and that the DbS model can parsimoniously capture these effects in a continuous and dynamical manner. These results were added in the Results section and presented in Figure 5—figure supplement 2 of the revised manuscript.

c) One possible reason why the two conditions did not lead to differences could be that – after doing one condition for two days – it might have been impossible for the participants to adjust their "habit-like" behavior to a new incentive rule. This could be checked by analyzing the first half of the task in a between-subject manner.

We agree with the reviewers that a possible reason why the two experimental conditions did not lead to differences could be that, after doing one condition for two days, the participants did not adapt as easily to the new incentive rule. However, note that as the participants did not know of the second condition before carrying it out, they could not adopt compromise strategies. Nevertheless, we fitted the latent-mixture model only to the first condition that was carried out by each participant. We found once again that DbS was the best model explaining the data, irrespective of condition and experimental paradigm. The results of these analyses are presented in Figure 4—figure supplement 7, and reported in the revised manuscript. Therefore, we conclude that the fact that DbS is favored in the results is not an artifact of carrying out two different conditions in the same participants.

3) Expository issues.a) Provide more intuition for the equivalence between results under different optimization criteria. Do any of these results rely on the asymptotics?

The exact equivalence that we obtain between the optimal coding rules for mutual information maximization and accuracy maximization depends on our use of an asymptotic approximation. As noted in our response to point 1a, though, we verify that the characterization of the optimal coding rule for accuracy maximization using the asymptotic approximation is not too different from what we find using a flexible numerical solution method in the case of small values of *n* (see Appendix 7). We do not examine the question of how closely the optimal coding rules under these two objectives continue to be similar in the case of small *n*, however. This is because our primary focus is on the difference between the optimal rule for accuracy maximization (when *m* can be unbounded) and DbS. We compare the optimal coding rule for accuracy maximization to the one that would maximize mutual information primarily in order to compare our results to other discussions in the efficient coding literature, and those results are all based on asymptotic approximations. Thus we too only consider the implications of mutual information maximization in the case of an asymptotic approximation of the kind commonly used in the prior literature.

b) The clarity of the Introduction can be improved. In the second paragraph, the authors suddenly jump to a discussion of differences between perceptual and preferential choice, but I think it would be more important to first make clear what the overall goal of the work is. The third paragraph is very confusing. Its first sentence is not even a full sentence (a verb is missing at "where only a finite number.…") and pretty much incomprehensible. Then, the work of Simon Laughlin is discussed, but it is questionable whether this is really the best way to motivate the proposal of a binary encoding system (why not simply saying that neurons provide binary outcomes). The reference to Query Theory in the fourth paragraph remains vague. In a later paragraph, the idea of adaptation to a frequency distribution is introduced without explaining what it actually means (and one reason for this is that DbS is not well explained in the previous paragraph).

We thank the reviewers for pointing this out. We agree that the Introduction in the initial submission was patchy and fragmented. We have now extensively rewritten the Introduction section, which we believe now better identifies the main focus of our work.

c) In the Abstract, the authors should make more clear what the task was about (though we understand that this isn't easy give the word limit). In addition, the word "Here" is used to start two consecutive sentences, and an "a" is missing at "strategy that might be utilized…".

We have rewritten the Abstract and also tried to incorporate the reviewers’ suggestion. Given the 150 words limit, it is difficult to describe the details of the task, but we have now made clear that it was a numerosity discrimination task.

4) Links to related literature.a) Do the results cast doubt on the argument made in the paper by Rustichini et al., 2017, which argues for a coding scheme based on expected utility maximization rather than mutual information?

Like Rustichini et al., we develop a theory of efficient coding that is based on optimization of the organism’s performance of a particular task (in our case, a discrimination task), rather than taking maximization of mutual information as the objective. (We do also provide some results about mutual information maximization, but this is for purposes of comparison of our theory to other discussions of efficient coding, which often assume that objective.) In this respect, our approach is fundamentally similar to theirs. There are however many differences between the particular problem that they address with their model and the situation that we analyze here. They consider a different decision task, a different class of possible encoding rules, and they also optimize their encoding and decoding rules for a particular prior distribution (that is, in terms of our formalism, they consider only the case in which *m* is unbounded); hence our results cannot be compared in detail to theirs. We find that our subjects do not behave much differently when we change the reward function used to incentivize their choices, and some may feel that this shows that the coding scheme that people use is not optimized to maximize reward. However, we only find that the coding scheme that would maximize expected reward is different in the case of the two incentive schemes when *m* is assumed to be unbounded; when it is instead important to economize on the number of samples from the prior that are used, we believe that DbS is equally efficient in either case, and this is the case that seems to be empirically relevant for our subjects. Thus we do not believe that our results contradict the Rustichini et al. position, though they cannot be said to prove that it must be correct, either.

b) Clarify that Equation 9 only corresponds to DbS in the asymptotic limit. The finite sample regime was emphasized in Bhui and Gershman, 2018, in order to explain certain phenomena (such as range effects) that do not follow directly from the CDF encoding function. Instead, that paper showed how these results could be obtained from a smoothed encoding function computed on a small set of samples. Relatedly, please clarify the links to the Bhui and Gershman paper. In particular, how does infomax in this paper (like in Supplementary Note 1) related to infomax in their paper? Also their work is described as adding noise after efficient coding, but this is not the case. The "noise" in that model comes purely from the fact that a finite number of samples are drawn, so that the sample-based CDF only approximates the true CDF.

This point is addressed in response to point 1a.

c) The authors should discuss (and scrutinize) their empirical findings a bit more in the context of other studies that have compared perceptual and preferential decisions, in particular Dutilh and Rieskamp, 2015, who studied a quite similar task (choosing based on the number of dots with a perceptual vs. a preferential incentive rule). Here, it was found that decisions were slowest for the most difficult trials in the perceptual condition but not in the preferential condition. The question is whether there are any interesting, related response time differences between the two conditions in the current task (because if the number of dots on the left and right is very similar, one should not think too long about it if the reward depends on the number of dots [preferential], but one would need to think for a long time about it to decipher which side has more dots [perceptual]).

The reviewers are right that recent studies have investigated behavior in tasks where perceptual and preferential decisions are also made in paradigms with identical visual stimuli. In these tasks, investigators have reported differences in behavior, in particular in the reaction times of the responses, possibly reflecting differences in behavioral strategies between perceptual and value-based decisions. Therefore, we investigated whether this was also the case in our data. We found that reaction times change as a function of the different performance metrics nearly as expected given the proportion of correct responses (Figure 4—figure supplement 5). However, we found that reaction times did not differ between experimental conditions for any of the different performance assessments considered here. While we agree that this result might appear surprising, it further supports the idea that subjects in our task were in fact using the DbS strategy irrespective of behavioral goals. These results are reported in the main text, and presented in Figure 4—figure supplement 5. We acknowledge that the study of efficiency in the context of free reaction time tasks is an interesting topic of research. However, we believe that it falls out of the scope of this study, but it is certainly a question that will be important to address in future work.

[Editors' note: further revisions were suggested prior to acceptance, as described below.]

Revisions for this paper:1) Having established the identifiability of the α parameter under the DbS model, it seems imperative to fit Experiment 3 using α as a free parameter and omitting the adaptation mechanism (this has been done in the revision but these results are used to examine whether there is adaptation or not, rather than to actually examine if the fitted prior differs between experiments 1-2 and 3). In other words, Figure 5A can be expanded to include the α fits from Experiment 3. This exercise can address whether there is indeed a change to the shape of the prior across experiments, which is a pivotal component of the proposed framework relative to alternative frameworks that assume complex representational non-linearities without sensitivity to the prior. The results in Figure 5C show that α in Experiment 3 converges to a lower asymptotic value. However, imposing an adaptation process, especially when there is no strong support for such process, can obscure the interpretation of the fits. Furthermore, I remain skeptical about the claim that there is dynamical adaptation within each experiment: i) if anything, the new analyses show that the "free α" model provides a better goodness of fit, and ii) Figures 4—figure supplement 2 and Figure 5—figure supplement 1 show no obvious dynamical trends in behavior. How does the adaptation manifest itself in the data? Figure 5—figure supplement 2 hints toward an early period in which the "free-α" fits worse than the dynamic α model (up to ~150 trials). Perhaps modeling this discrepancy explicitly (e.g. two α parameters for early and late trials, respectively) would suffice.

We followed the reviewer’s suggestion. In order to test whether the results obtained in Figure 5C were an artifact of the parametric assumption in the adaptation parameter, we ran a model using only the first 150 and last 350 trials of each daily session. The α parameter was allowed to vary between the first and last sets of daily trials and between Experiments 1–2 and Experiment 3. The results are shown (purple: Experiments 1-2, orange: Experiment 3). Each bar represents the mean value of the α parameter for a combination of experiments and set of daily trials. The α parameter is smaller at the end of Experiment 3 relative to the beginning of Experiments 1–2 and Experiment 3. In addition, the α parameter is lower in the last set of daily trials in Experiment 3 than in the last set of daily trials in Experiments 1–2. These results virtually reproduce the results obtained with the parametrization assumed in Figure 5C and therefore confirm that they are not artifacts of the parametric assumption. These results are now included in the new Figure 5D.

In order to allow a fair comparison with the results of the adaptation analyses presented in Figure 5C, in this model, we did not allow *n* to freely change for each condition. Therefore, a potential concern is that the results might be an artifact of changes in *n*, which could for example change with the engagement of the participants across the session. Given that we already demonstrated that both parameters *n* and *α* are identifiable, we fitted the same model as in Figure 5D, however this time we allowed *n* to be a free parameter alongside *α*. We found that the results obtained in Figure 5d remained virtually unchanged (see new Figure 5—figure supplement 3A), in addition to the result that the resource parameter *n* remained virtually identical across the whole session (see new Figure 5—figure supplement 3B). Altogether, these results provide further evidence of adaptation, highlighting the fact that the DbS model can parsimoniously capture adaptation to contextual changes in a continuous and dynamical manner. The results of these analyses are now presented in the revised manuscript